# Micromechanisms leading to shear failure of Opalinus Clay in a triaxial test: A high-resolution BIB-SEM study

Lisa Winhausen[1], Jop Klaver[2], Joyce Schmatz[2], Guillaume Desbois[3], Janos L. Urai[4], Florian Amann[1], Christoph Nussbaum[5]

5   [1] Department of Engineering Geology and Hydrogeology, RWTH Aachen University, 52064, Germany
[2] MaP - Microstructures and Pores GmbH, Lochnerstraße 4-20, 52064 Aachen, Germany
[3] Institute of Structural Geology, Tectonics and Geomechanics, RWTH Aachen University, 52064 Aachen, Germany
[4] Institute of Tectonics and Geodynamics, RWTH Aachen University, 52064 Aachen, Germany
[5] Federal Office of Topography Swisstopo, Route de la Gare 63, 2882 St.-Ursanne, Switzerland

10  *Correspondence to*: L. Winhausen (winhausen@lih.rwth-aachen.de)

**Abstract.** A microphysics-based understanding of mechanical and hydraulic processes in clay shales is required for developing advanced constitutive models, which can be extrapolated to long-term deformation. Although many geomechanical tests have been performed to characterise the bulk mechanical, hydro-mechanical and failure behaviour of Opalinus Clay, important questions remain about micromechanisms: How do microstructural evolution and deformation mechanisms control the complex rheology? What is the in-situ microstructural shear evolution and can it be mimicked in the laboratory? In this contribution, Scanning Electron Microscopy (SEM) was used to image microstructures in an Opalinus Clay sample deformed in an unconsolidated-undrained triaxial compression test at 4 MPa confining stress followed by Argon Broad Ion Beam (BIB) polishing. Axial load was applied (sub-) perpendicular to bedding until the sample failed. The test was terminated at an axial strain of 1.35%. Volumetric strain measurements showed bulk compaction throughout the compression test. Observations on the cm- to µm-scale showed that the samples exhibited shear failure and that deformation localised by forming a network of µm-wide fractures, which are oriented with angles of 50° with respect to horizontal. In BIB-SEM at the grain scale, macro-scale fractures are shown to be incipient shear bands, which show dilatant inter- and intragranular micro-fracturing, granular flow, bending of phyllosilicate grains, and pore collapse in fossils. Outside these zones, no deformation microstructures were observed indicating only localised permanent deformation. Thus, micromechanisms of deformation appear to be controlled by both brittle and ductile processes along preferred deformation bands. Anastomosing networks of fractures develop into the main deformation bands with widths up to tens of µm along which the sample fails. Microstructural observations and the stress-strain behaviour were integrated into a deformation model with three different stages of damage accumulation representative for the deformation of the compressed Opalinus Clay sample. Results on the microscale explain how the sample locally dilates while bulk measurement shows compaction, with an inferred major effect on permeability by an increase in hydraulic conductivity within the deformation band. Comparison with the microstructure of highly strained Opalinus Clay in fault zones shows partial similarity and suggests that during long-term deformation additional solution-precipitation processes operate.

# 1    Introduction

Due to its low permeability and self-sealing capability, Opalinus Clay (OPA) has been chosen as host rock for radioactive
waste disposal in deep geological formations in Switzerland. Major effort has been made to study its mechanical and hydro-mechanical behaviour in laboratory experiments (e.g. Nüesch, 1991; Aristorenas, 1992; Bock, 2010; Amann et al., 2011; Wild et al., 2015, 2018; Favero et al., 2018; Giger et al., 2018). From a geomechanical point of view, the behaviour of OPA is at the transition from a stiff soil to a weak rock. The strain response upon loading suggests a yielding stress threshold that coincides with the onset of dilation. Micro-acoustic measurements on both laboratory and field scale suggest that the onset of
dilatancy is associated with micro-acoustic events, which is typically observed for brittle rocks (Amann et al. 2011; Wild et al. 2015; Amann et al. 2018).

After sedimentation, physical compaction, and development of diagenetic bonding (Marschall et al., 2005), OPA shows a well-pronounced bedding and foliation resulting in transversely isotropic hydraulic and mechanical characteristics. Cm-scale lithological heterogeneity and the pronounced microfabric, i.e. preferred grain and pore orientation, govern the macroscopic
lamination. Houben et al. (2013, 2014) give an overview of the microstructure of the pristine material (unfaulted shaly OPA). SEM-based studies (scanning electron microscopy) have shown that the microfabric consists of larger sub-millimetre domains embedded in a matrix of subparallel platy clay aggregates (Klinkenberg, 2009; Houben et al., 2013, 2014; Seiphoori et al., 2017).

Typically, the mineral composition consists of the following components: Siderite, biogenic calcium carbonate (fossil shells),
mica, calcite, organic matter, quartz, pyrite, feldspar and clay matrix. These components can be porous such as the clay matrix, fossils, framboidal pyrite and organic matter or non-porous such as such as quartz-, mica- and calcite grains. The SEM-visible void space consists of bedding-parallel fractures, pores in the fossils or pyrite and three different pore types in the clay matrix (Houben et al., 2013; Desbois et al., 2009). The SEM-visible pore area can be described by a power-law distribution, which, when extrapolated to pore sizes at the resolution of the bulk porosity measurements, matches with the measured porosity of
the shaly facies of OPA of 15.3 % (Houben et al., 2013, 2014). This value is slightly lower than petrophysical porosities measured by various methods such as water content porosity, Helium-pycnometry and neutron scattering techniques (Busch et al., 2017). Elongated pores in the clay matrix, as well as elongated grains such as micas and calcite clasts are oriented parallel to bedding (Wenk et al., 2008; Houben et al., 2014). The combination of BIB-SEM (broad-ion beam-polishing and scanning electron microscopy), FIB-n (focused ion beam nano tomography) and STEM (scanning transmission electron microscopy)
allows a multi-scale description of the pore structure indicating a more connected pore network parallel to bedding as compared to normal to bedding (Keller et al., 2011, 2013).

Micromechanisms of deformation in clay-rich rocks are complex (e.g. Desbois et al., 2016; Chandler et al., 2016; Desbois et al., 2017; Forbes Inskip et al., 2018; Gehne et al., 2020; Nejati et al., 2020; Schuck et al., 2020). Early macroscopic studies on experimentally deformed shales by compressive loading show that failure is accompanied by shear faulting both across and
in-plane of anisotropy, plastic flow or slip along the plane of anisotropy, or kinking (McLamore and Gray, 1967). These

processes depend on the confining stress and the orientation of the anisotropy plane in respect to the applied stress. For example, deformation tests on shales from the Wilcox shale formation show that deformation at confining stresses less than 100 MPa is prone to develop macroscopic sharp shear fractures, while experiments at higher confining pressures above 100 MPa reveal macroscopic shear zones with en-enchelon fractures and kink bands following compositional layering (Ibanez and Kronenberg, 1993). Their microscale analyses indicate brittle, dilatant micro-fracturing over a wide range of scales where deformation is accompanied by kinking of clay minerals. The deformation behaviour of shales or clay-rich materials can be brittle, ductile or a combination of both. Underlying micromechanisms are cataclasis, granular flow, frictional sliding and crystal plasticity (Morgenstern and Tchalenko, 1967; Handin et al., 1963; Maltman, 1987; Logan et al., 1979; 1987; Wang et al., 1980; Lupini et al., 1981; Rutter et al., 1986; Blenkinsop, 2000; Desbois et al., 2016, Desbois et al., 2017; Schuck et al., 2020). Additionally, Ibanez and Kronenberg (1993) distinguished deformation modes based on the loading direction with respect to bedding. Individual and conjugate fractures with orientations between 33° to 73° towards horizontal were found in samples loaded normal to bedding, whereas samples loaded parallel to bedding are prone to develop macroscopic kink bands with orientation of 47° to 75° and material rotation. Shear zones with orientations of 56° to 46° including echelon fractures are typically observed for samples where maximum load is oriented by 45° degrees to bedding. The influence of anisotropy on fracture mode and propagation has also been studied in fracture toughness experiments on shales and shows that the direction of fracture growth is dependent on i) the orientation of anisotropy, ii) the extend of anisotropy, and ii) the loading mode (Forbes Inskip et al., 2018; Nejati et al., 2020).

Holland et al. (2006) presented a model for deformation of shales with a high Brittleness Index (Ingram and Urai, 1999) where initial deformation leads to the formation of dilatant micro-fracture arrays with strong increase in permeability, with progressive deformation and cataclasis of fragments forming a ductile clay gouge and resealing of the fractures.

Microphysical models of phyllosilicate-rich fault gouge for different crustal regimes have been developed using ring shear experiments with rock analogues consisting of mixtures of granular halite and fine-grained muscovite or kaolinite (Bos and Spiers, 2000, 2002). The rheology is controlled by frictional or frictional-viscous behaviour depending on strain rates and the effect of pressure solution. Later, the model was extended by plastic flow of phyllosilicates and the competing processes of shear-induced dilatation and compaction due to pressure solution (Niemeijer and Spiers, 2007).

For OPA, links from structural analyses to deformation mechanisms have been derived using natural outcrops and laboratory experiments. Faulted OPA in outcrops of the Jura mountains indicate discrete shear surfaces with thicknesses of $1 - 4$ µm (Jordan and Nüesch, 1991). These shiny and mirror-like surfaces (i.e. slickensides), optically and microscopically visible, consist of denser packed clay material. They are characterised by a grain size reduction and a preferred alignment of clay aggregates parallel to the shear zone forming R- and Y-shears (cf. Passchier and Trouw, 2005). Studies on the so-called 'Main Fault' in the Mont Terri Underground Research Laboratory (MT-URL) show that thin shear zones are constituted of slickensides on fracture surfaces along which samples break (Laurich et al. 2014, 2017). The dominantly brittle and friction-controlled deformation mechanism is supported by a series of compression tests covering a broad range of ultimate strains (ε up to >20 %) and confining stresses (50 to 250 MPa; Nüesch, 1989; Jordan and Nüesch, 1991). The authors interpreted that

the experimentally produced R-surfaces act as 'sliding surfaces' on water interlayers of clay minerals and comprise the dominant deformation mechanisms. Kaufhold et al. (2016) performed microstructural analyses using different X-ray computed tomography (CT) techniques with varying resolutions and found two prominent fracture sets in an experimentally deformed OPA under unconsolidated-undrained conditions at 6 MPa confining stress. Detailed investigations in combination with SEM show an obliquely oriented macro-fracture accompanied by small fractures, cracks and the rearrangement of particles. The

authors interpret the microstructural deformation as a mylonitic shear zone.

    Although CT and SEM analysis provide important insights for an advanced understanding of failure processes in OPA, they remain inconclusive at the grain scale. Currently, only few studies exist which investigate grain-scale processes using high-resolution images. Laurich et al. (2014, 2017, 2018) analysed the deformation structure from meso- down to grain scale on samples from the 'Main Fault' in the MT-URL. Results show a network of localised, micron-thick shear zones strongly reduced

in porosity, slickensides and scaly clay forming mostly undeformed microlithons. The formation of gouge is associated with both brittle deformation including micro-fracturing and cataclasis, pressure solution and ductile flow of the clay matrix forming relays which ultimately lead to the formation of scaly clay. The formation of scaly clay has also been analysed experimentally by use of direct shear tests on samples sheared both parallel and normal to bedding direction at various normal stresses (Orellana et al., 2018). Deformation is localised in R-shear planes creating slickenside surfaces and for samples sheared normal

to bedding a rotation of the original fabric creating millimetre-sized off-sets along the R-shears.

    However, there is at this point almost no data published on grain-scale deformation mechanisms and microstructure evolution of OPA in laboratory experiments. Furthermore, existing studies (cf. references above) present deformation structures resulting from shear displacements in the order of metres (fault rock) to centimetres (experimentally deformed samples) resulting in an advanced stage of deformation up to residual or ultimate rock strength. For understanding the rheology during the entire

deformation process, earlier stages of deformation and the influence of anisotropy on the grain scale are fundamental. In particular, the knowledge of micromechanical deformation and the localisation of strain are relevant for the implementation of damage in constitutive models (e.g. Oka et al., 1995; Pietruszszak, 1999; Kimoto et al., 2004; Haghighat and Pietruszczak, 2015).

    In this contribution, the deformation structures of a sample were analysed that has been deformed under triaxial stress

conditions to peak stress, where damage has been initiated but not reached its residual strength state. We integrate the bulk mechanical failure behaviour under well-defined experimental conditions with the underlying micro-deformation processes to constrain a microstructural progressive failure model. This work presents a first look and precedes a systematic study of multiple samples deformed at a range of conditions.

## 2 Material and methods

### 2.1 Material description and core sample preparation

A detailed description of the experimental setup, the petrophysical and mechanical results of the sample used in this study can be found in Amann et al. (2012, their sample #214-38) and will be summarised only briefly in this contribution. The core material, where the sample was extracted from, originated from shaly facies of Opalinus Clay of the Mont Terri Underground Research Laboratory. The typical mass fractions of the main mineralogical components are: (i) clay minerals (50-66%) composed of 2:1 layer with illite, muscovite, smectite, and illite-smectite mixed layer minerals (20-30%), chlorite (7-8%) and kaolinite (20-25%), (ii) quartz (10-20%), (iii) carbonates (8-20%), (iv) iron-rich minerals (4-6 %), and (v) feldspars (3-5%) (Thury and Bossart, 1999; Klinkenberg et al., 2009). The cores were drilled dry utilising triple-tube core barrels and compressed air-cooling followed by hermetical sealing in vacuum-evacuated foil (Amann et al., 2012). The cylindrical sample with a length of 179 mm and a diametre or 89 mm was prepared by dry cutting and polishing of end faces using a lathe with a bedding orientation inclined $85 \pm 5°$ to the sample axis. For the water content measurement and calculation of total porosity as well as saturation degree, remaining core pieces were used (ISRM, 1979; Amann et al., 2011) indicating full saturation for the sample used in this study.

### 2.2 Triaxial testing

The sample was installed in a triaxial testing machine equipped with two axial and one circumferential strain transducers. The unconsolidated, undrained triaxial compression test (UU test) was performed at constant 4 MPa confining stress and was circumferential-displacement-controlled to give a constant circumferential displacement rate of 0.08 mm/min (cf. Fairhurst and Hudson, 1999). The test was terminated right after peak stress showed an axial stress drop (Fig. 1). At the end of the test, a total circumferential strain of about -0.60% and a total axial strain of about 1.35% were achieved. At rupture, the axial stress was 15.3 MPa and the axial strain 1.25%. After the test, the sample remained hermetically sealed in the FEP Fluorocarbon jacket and was stored at room conditions.

### 2.3 Sample preparation for image analysis and BIB-SEM

For the microstructural analyses, the deformed sample was removed from the jacket, dried and mechanically stabilised with epoxy resin (Fig. 2). Subsequently, the sample was cut dry along the long axis of the sample and normal to the fracture plane using a diamond saw. Eight sub-samples along the fracture network were cut dry with a micro-diamond saw to a size of approximately 2 x 2 x 1 cm$^3$ (Fig. 3a). The surface of each sub-sample was mechanical pre-polished with SiC grinding papers from P800 to P4000 grit size and glued on a stainless steel sample holder with silver paste. Subsequently, the surfaces of the sub-samples were polished using the BIB surface polishing procedure (Leica TIC3X) using a low incident Ar-ion beam angle (10.5 ° at 7kV for 45 min followed by 4.5° at 5 KV for 4 hours and for some selected sub-samples followed by 4.5° at 5kV for 18 hours, similarly to e.g. Schuck et al., 2020; Laurich et al., 2018; Oelker, 2019).

The BIB-polished surfaces were imaged with a SEM (Zeiss Supra 55) after surface coating with a 7 nm thick tungsten layer (Leica EM ACE600). Sample areas imaged with SEM were mapped using mosaic imaging at magnifications ranging from x120 (pixel size of 2,440 nm) to x 30,000 (pixel size of 9.8 nm) depending on the total area and level of details desired. Single images were stitched automatically using the Aztec software (Oxford Instrument, version 2.3) to produce large mosaics. The typical strategy included imaging the total subsample area with BSE detector at low magnification (x110 – x200, 2,730 – 1,508 nm) to provide reference maps for localisation of sub-areas. Regions of interests were selected and imaged with BSE and SE2 detectors at magnifications ranging from x5,000 to x15,000 (61.8 – 20.1 nm) to provide meso-scale information about the mineral and pore fabrics. Finally, high-level microstructural details down to grain-and pore scale at magnifications up to x30,0000 (10.0 nm) are imaged with BSE and SE2 detectors to analyse mineral and pore structures. Pore segmentation and quantitative analyses were performed by applying threshold segmentation of SE2-images using the image processing software ImageJ.

## 3    Results

Prior to the qualitative description of macro- and microstructures encountered, we define the terminology for our observations and the subsequent discussion. First of all, we use – where possible and not explicitly implied in the section title – prefixes as 'macro' or 'micro' for structures, which are visible with the naked eye or under the SEM, i.e. under magnification of at least x100.  In general, we subdivide where possible the general term 'fracture' – the surface formed through an initially intact solid material – into 'tensile', 'shear' or 'hybrid fracture' according to their movement along the surface, i.e. normal, parallel or a mixture of both, respectively. Furthermore, we call broader zones of multiple fractures and deformed microstructure 'deformation band', in case for elongated zones with increased porosity 'dilation band', and where shearing is interpreted, we use the term 'shear band' (cf. Du Bernard 2002).

### 3.1    Macroscale observations

The deformed sample showed a fracture network with two fracture sets (Fig. 3). In the following, all angles are defined as deviations from the horizontal, i.e. minimum principal stress direction (see diagram in Fig. 3). The first set (marked in green and further referred to as fracture set 1) was oriented oblique to the horizontal by an average angle of 49.5° ($\Theta_1$), and the second fracture set was oriented sub-horizontal, i.e. parallel to the bedding, with an average angle of 9.8° ($\Theta_2$, marked in pink and further referred as fracture set 2).  At the top left corner, the sample showed a single fracture, which splits into branches at a distance of 2 cm from the top. Towards the tips of these branches, the fractures deflected sub-horizontally along bedding-parallel fractures. The inclination of the fracture set changed from approximately 40° of the single fracture at the top towards the branches by more than 50° in the centre of the sample with respect to the horizontal. The branching fractures (marked in green) displayed an anastomosing pattern forming crudely lens-shaped fragments. Additional oblique fractures sub-parallel to

the branching fracture were observed throughout the sample which are not connected to the central branching fracture but show relay connections between the two fracture sets (Fig. 3, black box).

## 3.2     Microscale observations

Selected regions for microscale observations were located close to the larger, macro-fractures of set 1 (Fig. 3). However, many of the long, high-aperture macro-fractures crossing larger parts of the sample were inferred to contain artefacts produced due

to unloading and dust accumulation during sawing and mechanical polishing. Additionally, some locations showed secondary gypsum mineralisation caused by chemical reactions during sample storage before or after the experiment, which was about five years. When exposed to air, pyrite will oxidise creating sulphate, and the presence of Ca-ions from calcite (and dolomite) will lead to the formation of gypsum. These mineral alterations are common features also observed in excavation-induced fractures of the EDZ (excavation damage zone; Vinsot et al., 2014; De Craen et al., 2008) and hence, the timing of this process

could not be determined. Therefore, these regions were excluded from microscale analysis.

The fractures of set 2 and the microstructure within their close vicinity, i.e. the mineral fabric, grain size and pore structure, corresponded to intact shaly OPA samples analysed by Houben et al. (2013, 2014). These bedding-parallel micro fractures were inferred to have formed in response to drying, preparation and/or unloading (Houben et al., 2013; Soe et al., 2009) and were therefore interpreted to have formed during and after the experiment. The obliquely oriented fractures of set 1, however,

correspond to localised deformation. In general, the deformation was presented in form of tensile micro-fractures or in form of elongated zones indicating shearing. Both cases comprised varying local dilatancy.

Micro-fractures, which belong to fracture set 1), were often oblique to bedding with angles in the range of 40° to 55° with respect to the horizontal. Their apertures were up to a few µm and they had variable lengths (Fig. 4). The fractures mainly formed in the clay matrix whereas adjacent, larger components, such as shell fragments and quartz grains, remained intact. In

a few cases, the fractures crossed larger, elongated mica grains producing intergranular fracturing or bending (Fig. 4(c),(d)). Locally, multiple dilatant fractures formed an anastomosing network with lens-shaped microlithons of up to 50 µm width (Fig. 5(a),(b)). The foliation within these microlithons, which is indicated by the shape-preferred orientation (SPO) of elongated grains and pores, is deflected towards the fractures. Some of these deformation structures showed less dilatancy, mostly ductile deformation and a more pronounced shear component on the grain scale (Fig. 5(c)). Here, the SPO of non-clay minerals and

clay aggregates was oblique with respect to the bedding and curved according to the shear sense of the micro-fracture. Additionally, thin mica grains were bent according to the shear sense or intersected by a fracture.

Apart from the micro-fractures and small strain regions, elongated deformation bands of variable widths up to tens of µm thickness and different deformation intensities were be observed. In these zones, structural deformation included a variety of brittle and ductile deformation markers: Disintegrated framboidal pyrite aggregates, broken and fragmented or collapsed fossil

shells, broken calcite and silica grains, inter- and transgranular fractures, bent and broken mica grains, delamination of clay aggregates and a change of foliation indicated by a rotated SPO of clay aggregates (Fig. 6). In the latter case, the original SPO

was broken up and elongated grains reoriented to a parallel alignment with the elongated deformation band and, hence, the shear movement.

The deformation bands indicated zones of increased porosity (cf. so-called "dilation bands" of Du Bernard et al, 2002) and were usually located in vicinity of the larger macro-fractures shown in Fig.3. Locally, narrower bands formed around large, angular and preserved matrix fragments with sizes up to 20 µm. The structure within these bands showed varying deformation intensities. Its characteristic microstructure shows individual grains separated from the surrounding, often indicated by porous rims, and clay aggregates as well as pores with a random distribution and in a loss of SPO (Fig.7). These bands with widths of up to 50 µm hosted a variety of deformation markers and showed an increased porosity in rims around larger quartz grains, by the break-up and disintegration of clay particles forming 'saddle reef pores' (Fig. 6,7(a),(b)), and by the disaggregation of collapsed fossil fragments and pyrite aggregates (Fig.7(c)). Additionally, they presented intra- and intergranular fractures, kinking of mica grains or a loss of SPO of clay aggregates and elongated pores. The transition between the deformed and undeformed zone appeared as a sharp boundary.

### 3.3    Pore size analysis

Qualitative observations of increased porosity within the deformation bands were verified by statistical pore size analysis. Figure 8 shows cumulative pore size distributions derived from binarised, segmented SE2-images within and outside of the deformation band for two different subsamples. The subsample locations are shown in Fig. 7 (blue boxes). Cumulative visible porosity in the undeformed zones were 2.84 and 4.07 % for subareas from subsamples SS#2 (Fig.8(a)) and SS#8 (Fig.8(b)) and for the deformed zones two subareas for each subsample yielded porosities of 16.13 and 21.16 %, and 8.79 % and 9.84%, respectively. Furthermore, undeformed zones are characterised by a larger portion of smaller pores, i.e. $\leq 0.006$ µm$^2$, and a minor portion of larger pores, i.e. $\geq 0.02$ µm$^2$, compared to the deformed zones.

## 4    Discussion

### 4.1    Deformation mechanisms

The two prominent macro-fracture sets (Fig.3) were formed due to different processes. Fracture set 1 is associated with shrinkage due to drying when the material is exposed to air. Bedding-parallel fracture formation is commonly found in clay-rich rocks at the micro- and macroscale (e.g. Soe et al., 2009; Ewy, 2015; Fauchille et al., 2016). Mechanical forces acting on the sample during preparation as well as oven-drying and storage between experiment and BIB-SEM analysis can promote artificial fractures. Analyses by Houben et al. (2013) showed that different drying techniques of OPA produces the same undeformed microstructure in between the unwanted artificial fractures. The undeformed microstructure in vicinity of these fractures was indistinguishable to the microstructure of undeformed OPA and is therefore not associated with damage in the triaxial experiment. The fracture set 2, however, is dominated by both brittle and ductile deformation. The general orientation

of the macro-fractures with respect to horizontal showed an average inclination of ca. 50° and was consistent with the orientation of micro-fractures (Fig. 4) and elongated deformation bands (Fig. 5,7) showing inclinations between 40° and 55°. Observations of the sample surface after the experiment revealed macroscopic oblique and bedding-parallel side-steps which were connected via relays (Amann et al., 2012). After drying and sample preparation for the SEM analysis, these side-steps appeared on the microscale as tensile fractures along the bedding, i.e. with inclinations of approximately 9° towards the horizontal, but could not be distinguished from bedding-parallel desiccation fractures which were ubiquitous within the sample. Indications of shearing along these sidesteps could not be inferred from our SEM analysis. Fracture kinking or deflection is a common process in anisotropic rocks and is due to the competing influences of applied load direction and the direction of anisotropy plane (Forbes Inskip et al., 2018; Nejati et al., 2020). Our observations suggest that both factors play a role for fracture orientation and propagation since, on the grain scale, fractures cross the bedding and form along bedding (Fig.4a, Fig. 5). However, more work on samples with various loading configurations is required to quantify the influence of mechanical anisotropy on the fracture orientation.

In general, the deformation of both the clay-rich matrix and larger quartz, calcite and mica grains as well as pyrite aggregates is governed by brittle or ductile processes. On the one hand, grain scale clay splitting, grain abrasion and intra- and intergranular fracturing points to cataclasis localised in the deformation band. On the other hand, reorientation of clay aggregates and bending of elongated grains indicate structural reworking on grain scale to enable granular sliding. The former processes suggest a local loss of cohesion whereas the latter process enables the development of a new foliation parallel to the shear direction. The rock rheology is thus dominated by mixed-mode deformation due to the distributed contrasting stiffness of the multi-component mineralogy. Besides many other factors controlling the bulk failure behaviour, e.g. the amount of effective stress, the orientation of applied stresses, temperature and deformation rate (Popp and Salzer, 2006; Giger et al., 2018; Wild and Amann, 2018), our microstructural observations on OPA promote an intermediate tendency to brittle behaviour compared to other clay-rich rocks, e.g. the Callovo-Oxfordian mudrock (COx) from France and the Boom Clay (BC) from Belgium. The deformation mechanism of COx under triaxial compression is mainly associated with cataclastic mechanisms by grain and matrix fracturing, abrasion and comminution (Desbois et al., 2017). In contrast, deformation mechanisms in BC consist of pore collapse outside of the shear zone and grain boundary sliding as well as particle rotation within the shear zone (Oelker, 2019; Schuck et al., 2020). We note here that these experiments were conducted up to axial strains of ca. 6 and 20 %, respectively. The deformation structures and inferred mechanisms can be related to the amount of calcite, which serves as the cementing agent in these rocks (Klinkenberg et al., 2009; Kaufhold et al., 2013). For OPA, a large amount of calcite is part of the fossil shells rather than diagenetic bonding.

Deformation bands showed an increased internal porosity compared to the surrounding microstructure (Fig. 7). The abundant intra- and intergranular fractures, the delamination of clay aggregates creating 'saddle-reef pore' structures, i.e. the pore space created at the hinge of a kinked or folded mineral (cf. Hildenbrand et al., 2005), and the strained and collapsed fossil fragments within these dilated zones created space required for the particle movement and led to a locally increased porosity. Porosity values based on SE2-image segmentation of different subareas were 4 to 5 times higher inside these deformation bands

compared to the intact rock (Fig. 8). The pore size distribution showed a higher amount of smaller pores in the undeformed zone and a higher amount of larger pores within the deformed zone. We note here that the estimated porosity may be underestimated due to the resolution of our SEM-images. However, Houben et al. (2014) showed that porosity upscaling from a power-law distribution function of pore sizes yields porosity values, which are in the range of values determined by e.g.

water loss porosity measurements. Our estimations on visible porosity outside of the deformation bands are in agreement to that of undeformed OPA from the shaly facies of MT-URL (Houben et al., 2014). On the other hand, the porosity may be overestimated as part of it could have formed during unloading of the sample. If we assume that during unloading the compressibility is roughly equal in and outside deformation bands, then the ratio of porosity within and outside the deformation bands remains constant. Keeping all the above in mind, we infer that the deformation bands formed a local increase of porosity

during the experiment.

## 4.2    Damage accumulation during compressive loading

In our triaxial experiment, we observe that elastic and irreversible inelastic deformation occurred, and that both occurred simultaneously. Elastic deformation took place by deformation of the solid component and of pores – this was inferred to be

reversed by unloading the sample. From our SEM investigations, plastic deformation was governed by damage accumulation in deformation bands, but also by homogeneous matrix bulk compaction (cf. Allirot et al., 1977). Bulk compression in clay-rich rocks is associated with clay matrix compaction and pore collapse (Schuck et al., 2020). However, matrix compaction is difficult to image as a large portion of pores in the clay matrix is at the nanometre scale (Houben et al., 2014; Hemes et al., 2016; Klaver et al., 2015), which cannot be resolved using a SEM. For the case of localised plastic deformation, the

microstructural analysis showed a local damage accumulation in form of i) µm-thin micro-fractures (fracture set 1) with a strong preferred orientation of 50° towards horizontal (Fig. 3), and ii) wider deformation bands with increased porosity. The deformation markers (Fig. 4-7) are inferred to have formed due to shear failure since these are absent in the microstructure of tectonically undeformed OPA samples from MT-URL (Houben et al., 2013, 2014). Furthermore, Winhausen et al. (2021) analysed the microstructure of a shaly OPA sample, which has been used for hydraulic testing at maximum 24 MPa effective

stress, and found no deformation features. Their comparison to a non-loaded twin-sample as well as to other microstructural studies on OPA indicate no structural damages due to isostatic loading at this effective stress (Winhausen et al., 2021). The formation of loading-parallel, dilatant micro-fractures as response to axial compression is a well-known in brittle-deforming, hard rocks (e.g. Bieniawski, 1967; Scholz, 1968; Tapponier and Brace, 1976). Similarly, micro-fracturing during compressive deformation has also been demonstrated for OPA by micro-acoustic emission (Amann et al., 2012) as well as P-

and S- wave velocity measurements (Popp and Salzer, 2007). Amann et al. (2012) showed that fracture initiation, the point where the volumetric strain curve deviated from linearity ("crack initiation" in Amann et al., 2012), occurs long before stress peak at approximately 2 MPa differential stress independent from confining stress. Corkum and Martin (2007) discussed

uniaxial and triaxial test on OPA and presented similar results. They associated the opening of micro-fractures with damage by the breaking of diagenetic bonds in tensile failure, which ultimately leads to a degradation in stiffness and strength.

However, the orientation and distribution of micro-fractures remained unknown. Based on our microstructural analysis, the concept of micro-fracturing can be applied in slightly modified form compared to the theory previously formulated for hard, brittle rocks: Tensile microfractures form as obliquely – instead of vertically – oriented fractures along grain boundaries, which is inferred to be a result of loading direction and the orientation of anisotropy plane. This leads to a non-linear increase of bulk radial extension and a simultaneous reduction in stiffness based on the axial strain response. The process of fracture nucleation

coincides with the onset of volumetric dilation, i.e. the deviation from linearity of the bulk volumetric strain (Amann et. al, 2011), and it is interpreted as a non-localised process within the sample as reported in other studies (Kranz, 1983; and references therein).

Towards an advanced deformation stage close to the stress peak, these fractures localise along distinct planes within the sample and ultimately develop to macroscopic shear bands as observed on both hard rocks (Lockner et al., 1992; Tapponier and Brace,

1976) and other shales (Sarout et al., 2017). Our microstructural analysis showed an increased number of micro-fractures in vicinity of the macroscopic shear fracture, showing the extent of localisation. Furthermore, anastomosing fracture networks (Fig. 4) support the assumption that the single tensile-mode fractures formed in pre-failure. With progressive deformation, they changed into hybrid-mode with a shear component and coalesced to develop larger deformation bands or, once fully developed, shear zones. The deformation bands are therefore interpreted to have formed from multiple, coalescing micro-

fractures as supported by the fact that some immature bands still contain undeformed microlithons of the host rock. Adjacent tips of micro-fractures are joined by both brittle and ductile deformation mechanisms on the grain scale. These anastomosing fracture networks also show the influence of mechanical anisotropy as dilatant fractures form along the bedding plane as opposed to the general orientation of the fracture network (Fig.6(a)).

Once these deformation bands were established, shear strain localised within them and the internal structure became disrupted.

Shear movement along the deformation band boundary could be inferred from bending of clay particles (Fig. 5, 6(c), 8) and the highly strained and stretched fossil shell fragments (Fig. 8(c') and 8(c'')). Furthermore, the internal microstructure within these bands, i.e. the porous rims, the fractures around and through larger particles, and the random distribution of such, indicated cataclastic flow by grain rotation and material destruction. Correlation indicators for absolute shear displacement within and along these zones were missing. However, if strain would have localised within and along these bands, estimations

derived from the amount of maximum principal strains during the test and the orientation of the deformation bands would yield in an accommodation of maximum 0.4 % shear strain along these zones.

At the confining stress tested, OPA from the MT-URL typically shows a brittle bulk mechanical behaviour with ongoing deformation beyond peak stress resulting in a stress decrease to residual strength (Wild and Amann, 2018; Giger et al., 2018; Favero et al., 2018). The deformation structures presented in this study support the strain softening behaviour and a reducing

frictional resistance: Cohesional bonds were partly lost within the observed deformation bands by intergranular fractures and grain rotation, and clay particle rotation led to an alignment parallel to the deformation band enhancing clay-particle sliding.

In this regime, we hypothesise that the deformation bands would continuously be sheared and form a macroscopic shear band through the sample in the post peak region (Morgenstern and Tchalenko, 1967).

Cataclastic flow and particle movement within these deformation bands, led to a local porosity increase at the confining stress tested. On the other hand, the bulk mechanical behaviour showed a net increase of volumetric strain, i.e. bulk compaction, up and shortly beyond peak stress (Fig.1). This indicates that the matrix compaction was dominating the bulk deformation process to this stage. At the same time, a local increase of porosity may imply an increased permeability. This is striking and important to note since µm-thin shear zones from the 'Main Fault' in OPA at the MT-URL usually present a decreased porosity (Laurich et al., 2017) and deformed OPA in direct-shear experiments shows a reduction in permeability (Bakker and Bresser, 2020). However, fracture-sealing veins (mostly calcite) in the 'Main Fault' also indicate a paleo-fluid flow. Isotopic studies suggest that these veins formed contemporaneously with the initial fault activation, which supports or results in localised dilation in the early stage of shear development (Clauer et al., 2017).

The local increase of porosity in the deformation band leads to a local reduction in pore water pressure under undrained conditions and a related local increase in effective stress in the shear zones. The increase in effective stress involves a shear hardening effect, which represents the competing effect of shear softening due to the loss of cohesional bonds. Here, further research is required to investigate the dependency of permeability and porosity evolution with increasing amount of shear stress and the effect of undrained and drained conditions within the sample.

Some structures found in this study, such as the anastomosing fracture network, strained fossil shells and bending of phylosillicates resemble those found in highly strained OPA (Laurich et al., 2017; Orellana et al., 2018) and predict the early stage of Y-shear development. However, highly sheared OPA is characterised by grain size and porosity reduction, pressure solution and gouge development (Laurich et al., 2017). Similar observations have been made on other clay-rich rocks (e.g. Haines et al., 2013; Holland et al., 2006; Rutter et al., 1986). Therefore, microstructures and related processes investigated on a sample, which has been experimentally deformed up to peak-stress, resemble partly those observed in naturally deformed OPA. Furthermore, the influence of strain amount and strain rate, pore pressure evolution during deformation, pressure solution and mineral precipitation as well as self-sealing plays an important role (Laurich et al., 2018; Voltolini and Ajo-Franklin, 2020).

Many of these processes become effective at long time scales under natural conditions. In tunnelling however, low-permeable rocks such as OPA are subjected to undrained loading conditions during excavation. Thus, the triaxial compression tests have been performed as classical UU tests without prior saturation and consolidation. The pore water pressure development and effective stress inside the sample remain unknown. The test provide the undrained shear strength representative for the nearfield of a tunnel excavated at the consolidation conditions found at the MT-URL (Martin and Lanyon, 2003). Therefore, micro-deformation processes observed in the sample are considered similar to those found in the near-field of a freshly excavated tunnel in Mont Terri. This is further corroborated by the similarities between microstructural EDZ and BDZ (borehole damage zone) observations (Bossart et al., 2002, 2004; Yong et al., 2007; Nussbaum et al., 2011) and microstructures found in this study.

These studies show that purely extensional fractures coexist with shear fractures in the EDZ/BDZ. The latter – compared to tectonic fractures – are characterised by poorly developed striations but indicate re-oriented clay particles (Nussbaum et al., 2011). Furthermore, BDZ structures in a resin-impregnated over-core from the shaly facies of OPA at MT-URL showed branching fracture patterns on a mm-scale similar to the structures observed in our study (Kupferschmid et al., 2015).

Studies on samples deformed under undrained consolidated conditions to higher strains are currently under investigation and will address the effects of pore pressure, effective stress conditions, and sample orientation on the development of deformation microstructures.

## 4.3    Microstructural deformation model

For the unconsolidated, undrained deformation of OPA under differential compressive load at a total confinement of 4 MPa,
we propose the following deformation mechanisms based on our microstructural observations in combination with the bulk mechanical behaviour (Fig. 9): In the early stage of differential loading, the sample is subjected to axial shortening and elastic compression up to a differential stress of approximately 2 MPa. From stage (1), yielding starts caused by the formation of fractures and the related reduction of stiffness. These fractures form in the whole sample with preferred orientations crossing the bedding and with ongoing deformation, they concentrate with a higher density in the centre. The non-linear increase of
volumetric strain indicates a more abundant fracture formation in the beginning at lower axial strains. At stage (2), these fractures coalesce to create networks forming deformation zones with widths in the range of 20 to 50 µm. Clay bending and intracrystalline fractures correspond to a coexistence of brittle and ductile deformation processes on the grain scale due to the contrasting mineral rheology. When peak stress is reached and slightly exceeded at stage (3), shearing initiates and deformation bands begin to develop with distinct boundaries to the undeformed host rock. These deformation bands are characterised by
both brittle and ductile deformation in form of matrix cataclasis and reorientation of mineral fabric on the µm to tens of µm scale, even though on a bulk scale the sample deforms in a brittle manner. Locally, these zones showed increased porosity due to strained and crushed fossil shells, rotated calcite and quartz clasts, and by delamination of clay particles forming 'saddle-reef pores' (cf. Fig6, Fig9 (3a)). We note here that the microstructure in vicinity of the larger macro-fractures, i.e. close to the 'main' fractures, where potentially most of the shear strain is accommodated, host microstructural artefacts from sample
preparation (cf. section 3.2) and we hypothesise that with ongoing deformation shear strain will localise along a macroscopic shear band cross-cutting the sample with a reduction in porosity and grain sizes.

## 5    Implications and conclusions

We derived a deformation model for Opalinus Clay based on a sample that has been deformed in a triaxial test and whose post-experimental microstructure was analysed using BIB-SEM. The strain-controlled test (circumferential-displacement-control
rate of 0.08 mm/min) was performed at 4 MPa confining stress under undrained and unconsolidated conditions. The bulk mechanical behaviour was controlled by the microscale deformation, which is a combination of brittle and ductile processes.

The microstructural damage in OPA consists of, with respect to the horizontal, obliquely oriented single and coalescing micro-fractures as well as up to 50 µm-thick deformation bands pointing to deformation processes such as pore collapse, grain rotation and the rearrangement of clay aggregates due to bending and shear straining. Even though the pre-peak bulk volumetric strain showed a compacting behaviour, the microstructure was characterised by dilatant fractures and a local increase in porosity in the deformation bands, i.e. incipient shear bands. Consequently, the accumulation of damage upon compression created additional pathways for fluid flow, which can increase the permeability of the rock.

Our high-resolution study gives insights into the macro- and microscale structures with some similarities to excavation or naturally induced structures. However, OPA - deformed up to peak stress - also shows structural differences to naturally deformed OPA, which shows drastically reduced porosity. Strain amount and rate in natural fault systems, as well as elevated temperatures or clay-mineral hydration processes may influence the deformation mechanisms and structures. On the other hand, natural veins as an indication for paleo-fluid flow in the early faulting stage (Clauer et al., 2017) support localised dilation as similarly found in this study. Future experiments covering a broader range of boundary conditions on consolidated samples are required to analyse the deformation processes under shallow upper crustal conditions, and to formulate an effective stress dependent deformation model.

The results of this study support recent efforts (e.g. Pardoen et al., 2020; Jameei and Pietruszczak, 2021) to provide input to constrain damage in constitutive models of OPA or similar clay shales, which allow reproducing local porosity and permeability changes due to fracture formation, fracture coalescence, and the development of shear zones. Further studies on samples tested to larger axial strains on consolidated-undrained samples will test our model of strain localisation and pore space reduction. Furthermore, this will include the role of pore pressure development during deformation, and it will offer the possibility to properly compare experimentally and naturally deformed OPA.

*Data availability.* High-resolution BIB-SEM images and pore size data are available in the supplement provided.

*Competing interests.* The authors declare that they have no conflict of interest.

*Author contributions.* GD and JK performed sample preparation and BIB-SEM microscopy. JLU acquired funding, managed the project and discussed results at all stages. FA carried out the deformation experiment, acquired funding and managed the project. LW and GD analysed the data and LW wrote the text with contributions from all co-authors. All co-authors contributed to the discussion.

*Acknowledgements.* We would like to thank Swisstopo for the financial support. Ben Laurich and two anonymous reviewers are thanked for their thorough reviews and constructive comments, which helped to improve the manuscript significantly.

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

## 7    FIGURES

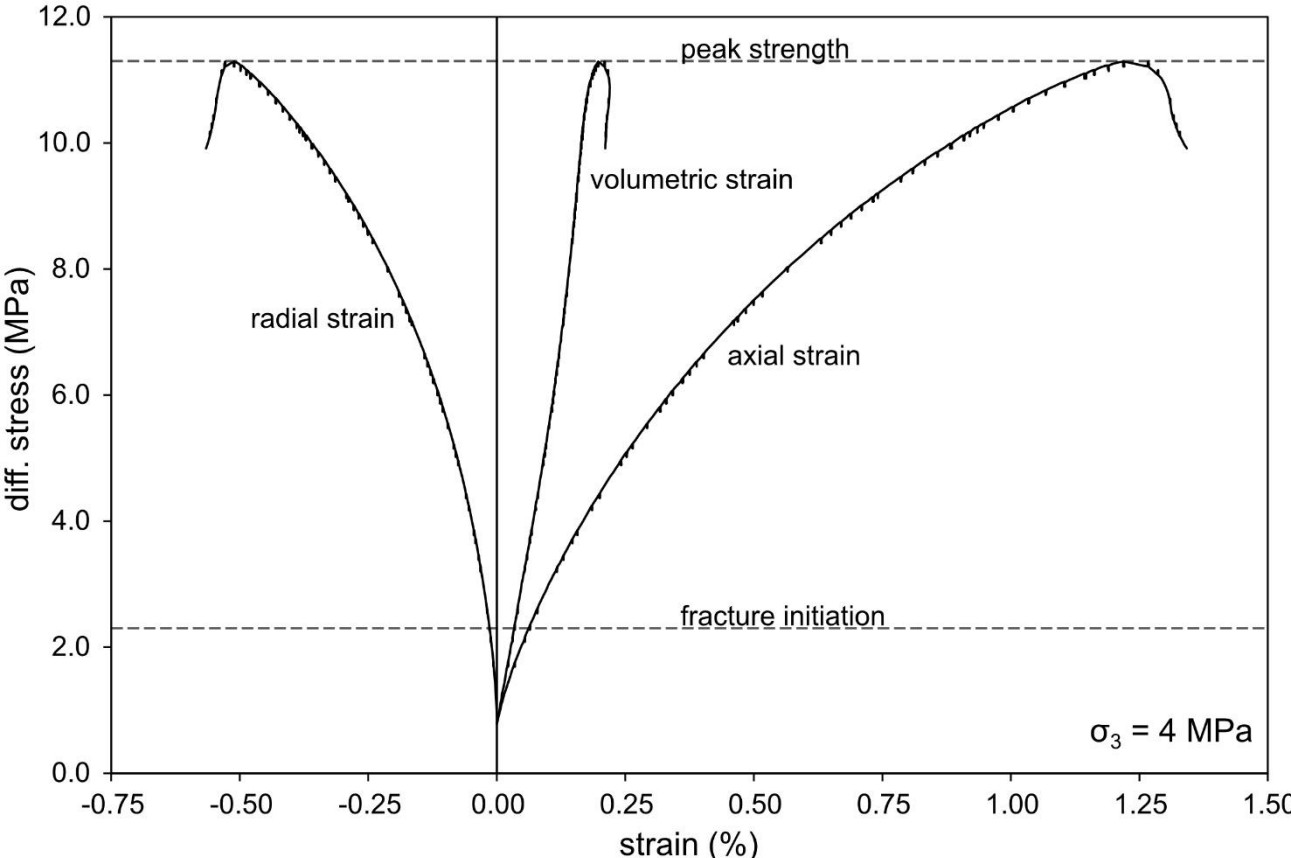

Figure 1: Stress versus strain development during the shearing phase. The shearing was conducted circumferential-displacement-controlled (0.08 mm/min) and at constant confining stress of 4 MPa. Peak stress was reached at 11.3 MPa differential stress and fracture (cf. Amann et al. 2012) initiation was observed at 2.3 MPa. The volumetric strain curve corresponds to bulk compaction over the entire experiment.


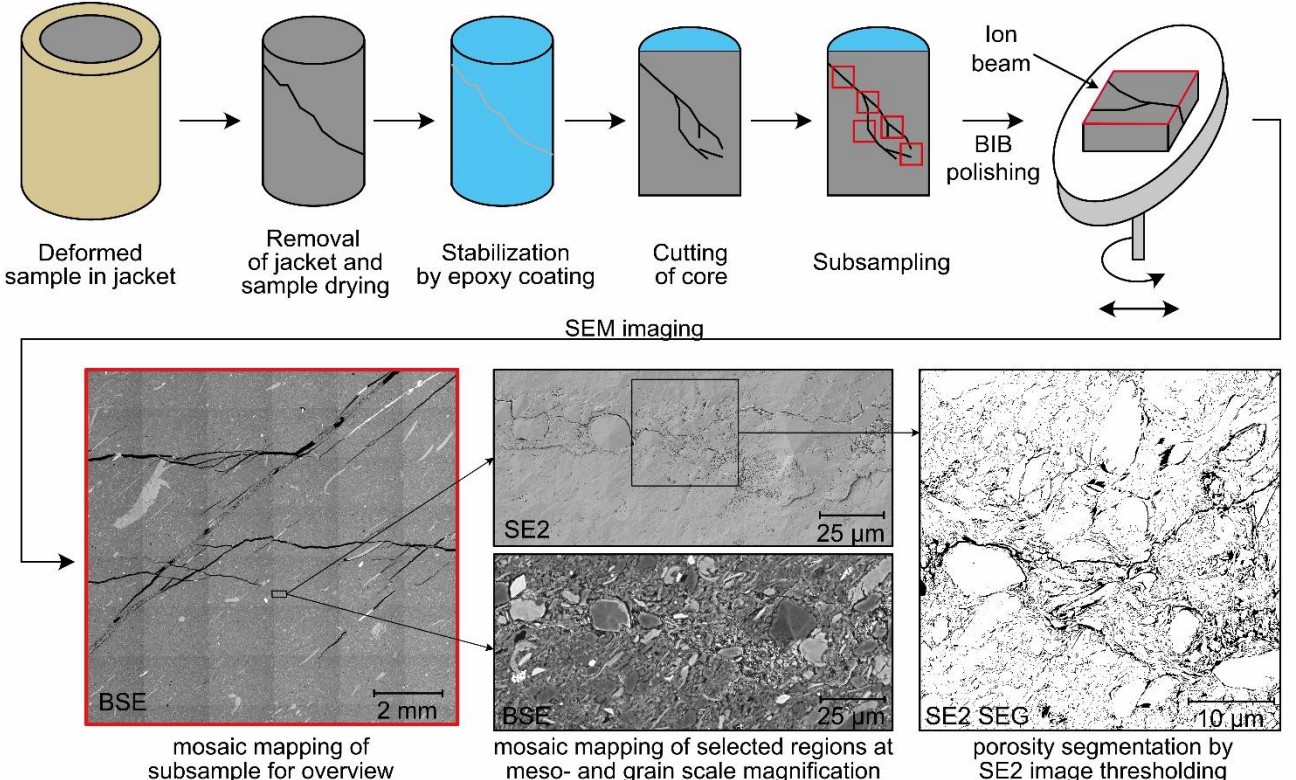

**Figure 2: Methodology and sample preparation for the BIB-SEM:** After dismantling and sample stabilisation by epoxy resin, the sample was cut normal to the fracture plane. Subsampling followed on selected regions of interest (red boxes), which were subjected to ion beam polishing. Finally, BSE-SEM micrographs were taken at low magnifications (110x to 200x) and detailed high-resolution SE2- and BSE-SEM images were produced from regions of interest. Pore segmentation was carried out by threshold segmentation of SE2 images.


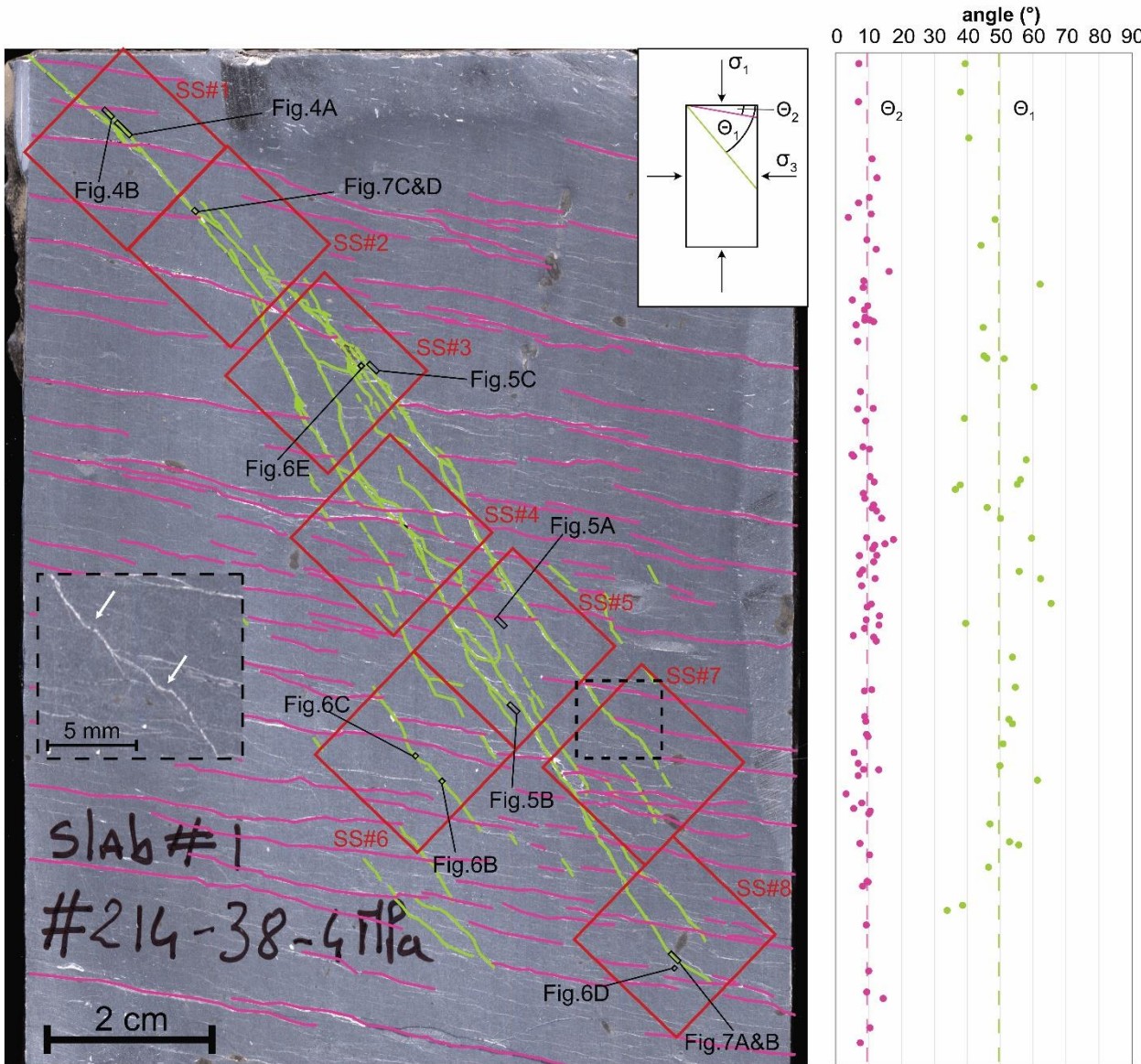

**Figure 3:** Macroscopic observations of the deformed sample. Red boxes indicate the location of the subsamples and black boxes show the selected regions of interest for large mosaic maps at meso- and grain scale magnifications. The macroscopically mapped fracture network showed two fracture sets: The fracture set 1 (green) showed a branching pattern and had an average inclination angle of 49.5° ($\Theta_1$), whereas the sub-horizontal fractures of fracture set 2 (pink) were inclined by 9.8° ($\Theta_2$) in respect to horizontal. Angles of mapped fracture set 1 along the sample length showed a slight increase towards the centre of the sample; the angles of fracture set 2 were consistent throughout the entire sample.

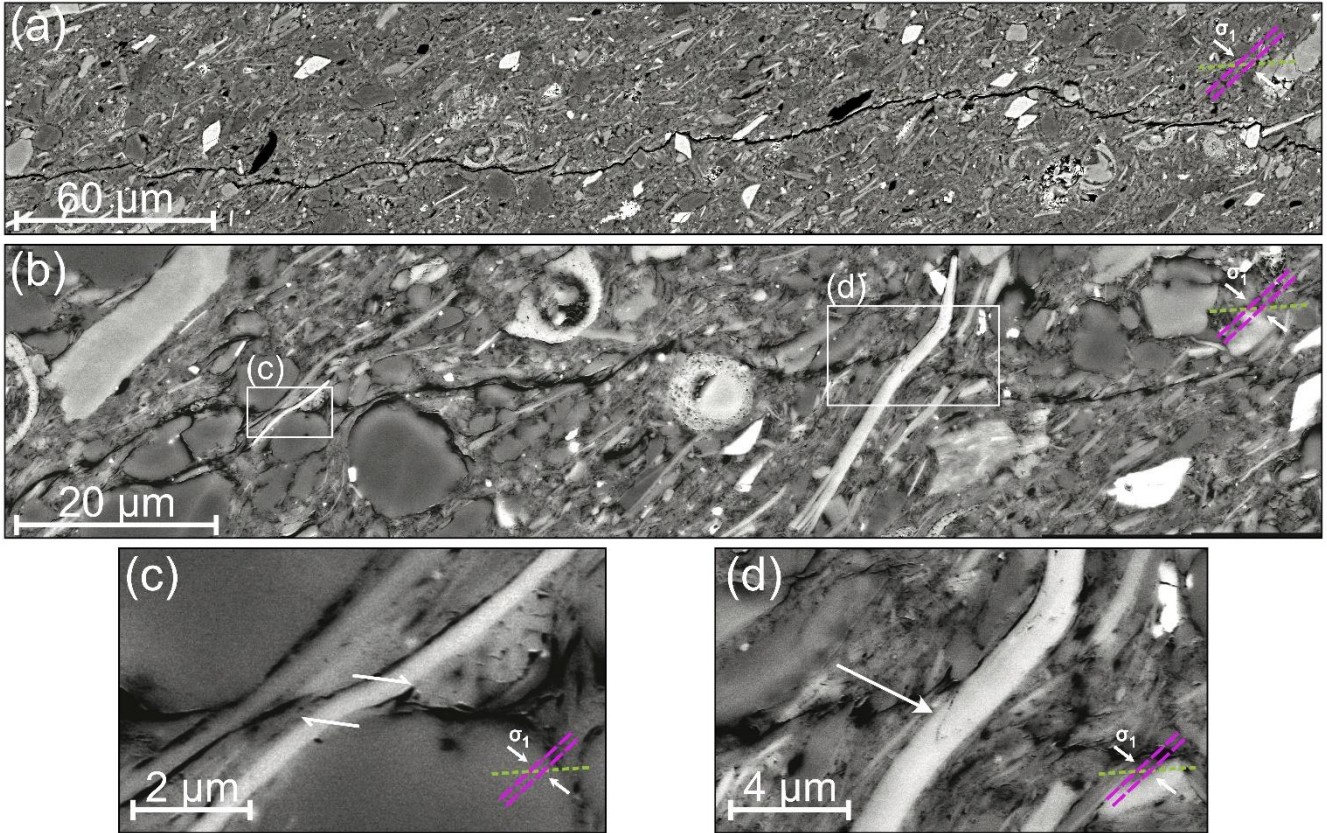

**Figure 4: BIB-SEM micrographs (BSE) showing micro-fractures of variable sizes with apertures of a few μm, which cross the bedding obliquely. Fractures were forming along larger grains (a) or across elongated mica grains (b) by either creating shear displacements (c) or brittle transgranular fractures (d). Maximum loading direction as well as macroscopic fracture set 1 (green) and bedding direction (pink) are indicated for each micrograph in this and all subsequent figures.**


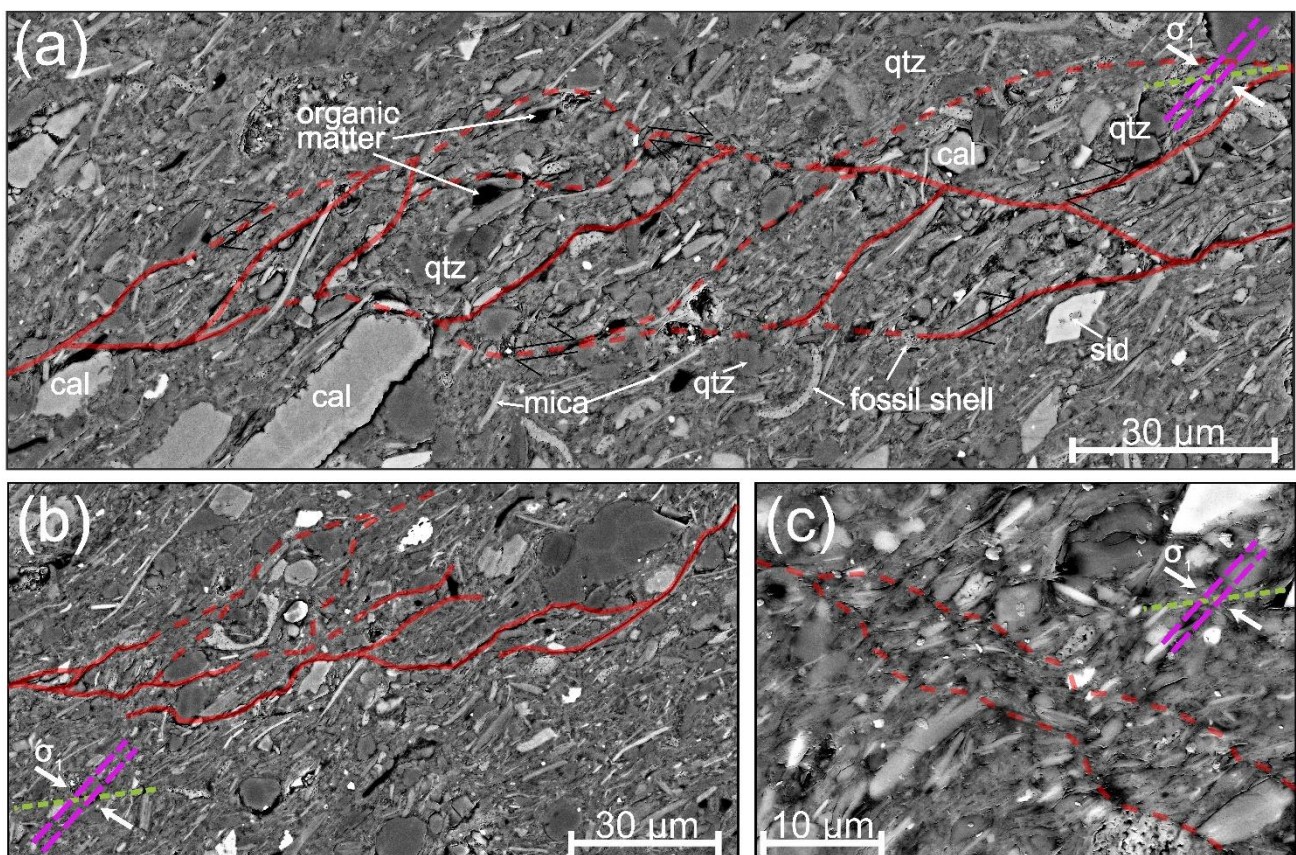

**Figure 5: BIB-SEM (BSE-) micrographs showing typical deformation structures associated with both brittle and ductile deformation. Red lines indicate strain localisation in form of fractures (a,b) or lineations of shear displacement (c). (a) anastomosing intergranular fractures with apertures of some µm formed lense-shaped islands whereas some grains displayed intragranular fractures. (b) shows fractures with intergranular porosity, which formed a not fully-developed network with minor indications of ductile deformation by clay-layer bending. (c) shows a deformation band with low porosity and predominantly ductile deformation indicated by the change of particle alignments.**



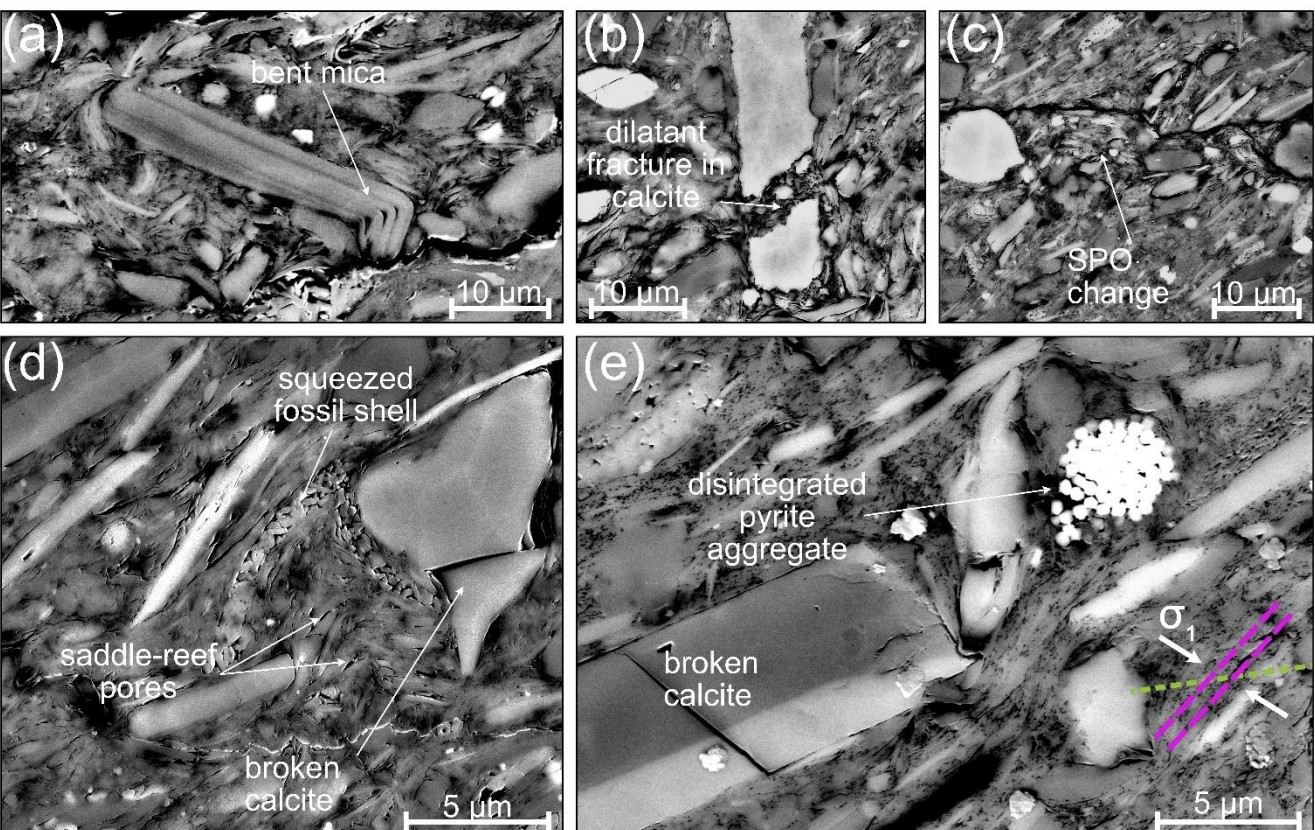

**Figure 6: BIB-SEM (BSE-)micrographs presenting different deformation markers: Bent and delaminated mica grain (a), dilatant fracturing of calcite grain and matrix fill (b), loss of shape-preferred orientation (SPO) of clay aggregates and elongated minerals parallel to the direction of displacement (c), squeezed and broken fossil shell, broken calcite grain and 'saddle-reef pores' (d), broken calcite grain and disintegrated pyrite aggregate (e). Illustration of loading direction as well as macroscopic fracture (green) and bedding direction (pink) in (e) applies for all micrographs in this figure.**

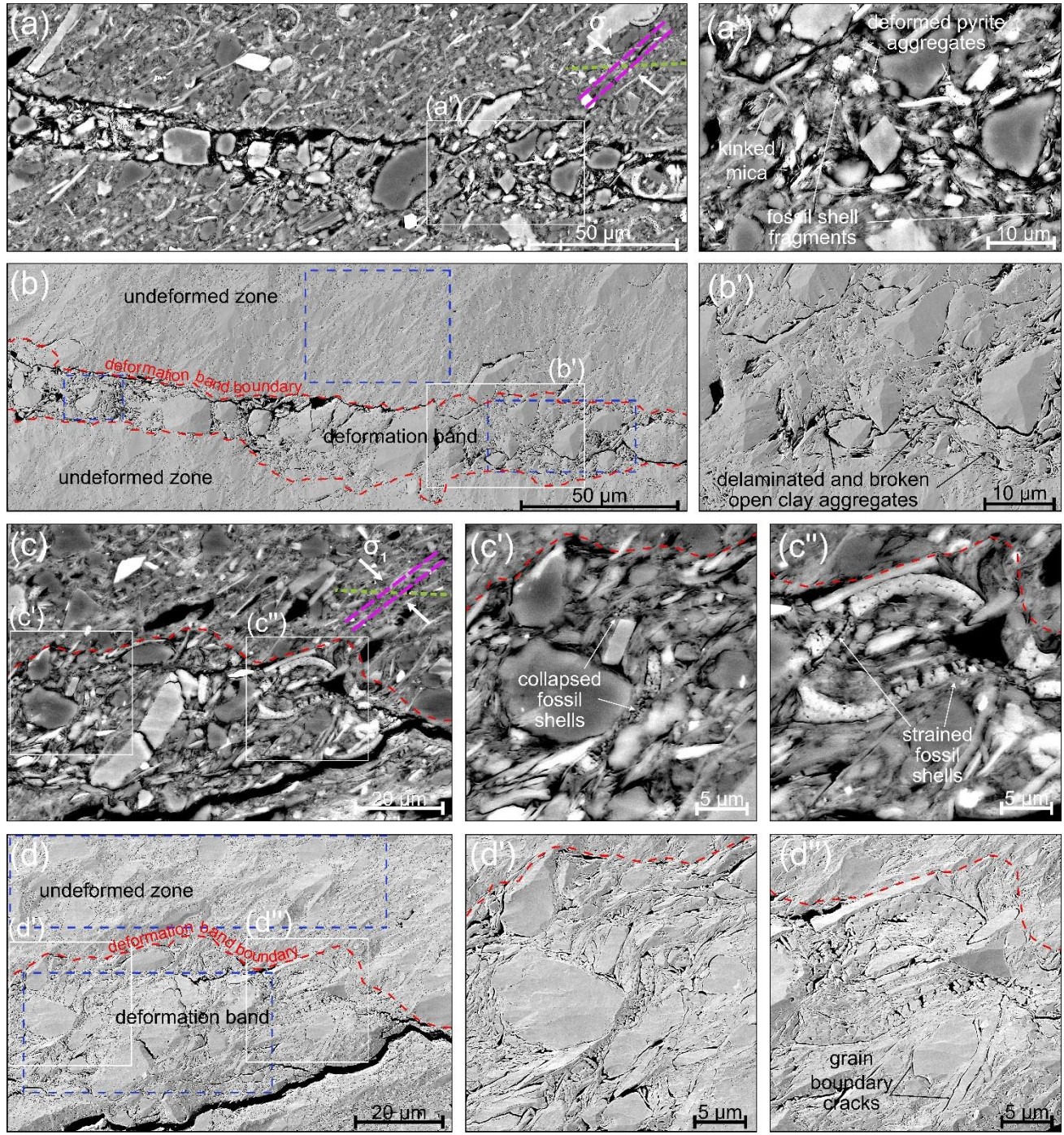

**Figure 7: BIB-SEM micrographs showing a zone of increased porosity. (a, a') and (c, c',c'') are BSE-images indicating the grain structures within the dilatantion band of up to 25 μm thickness. The detailed micrograph (a') shows typical strain indicators such as kinked mica, deformed pyrite aggregates and remnants of fossil shells. (b) and (b') are SE2-images showing the pore structure with considerably increased porosity along and within the deformation band, and the delamination of clay aggregates at higher**

magnification. (c) and (c',c'') present intragranular fractures and strained as well as collapsed and disintegrated fossil shells. SE2-image (d) shows an increased number of fractures within the deformation band, as well as increased pore space in between disintegrated shell fragments and clay aggregates (d'). (d'') displays grain boundary fractures, which appear as porous rims around quartz grains. Blue boxes indicate the locations where porosity has been measured.

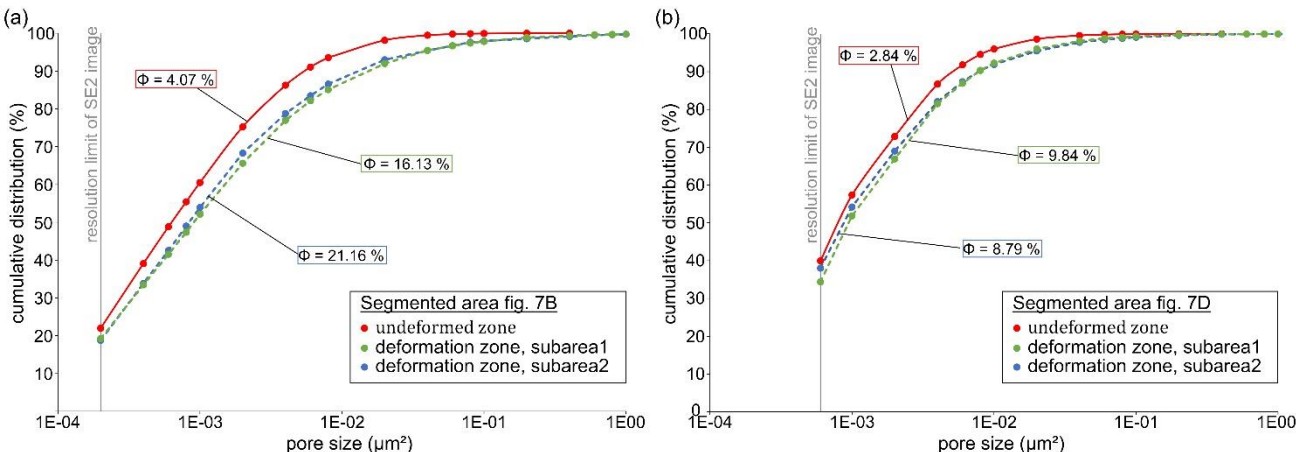

**Figure 8: Pore size distributions of undeformed and deformed OPA based on SE2-image segmentation from Fig. 7(b) and 7(d). Deformed zones present an increased porosity by a factor of 4 to 5, and show a higher portion of larger pores. Note that porosities estimated from SEM-image segmentation are underestimated due to the limited image resolution of 2E-04 $\mu m^2$ and 6E-04 $\mu m^2$, respectively.**

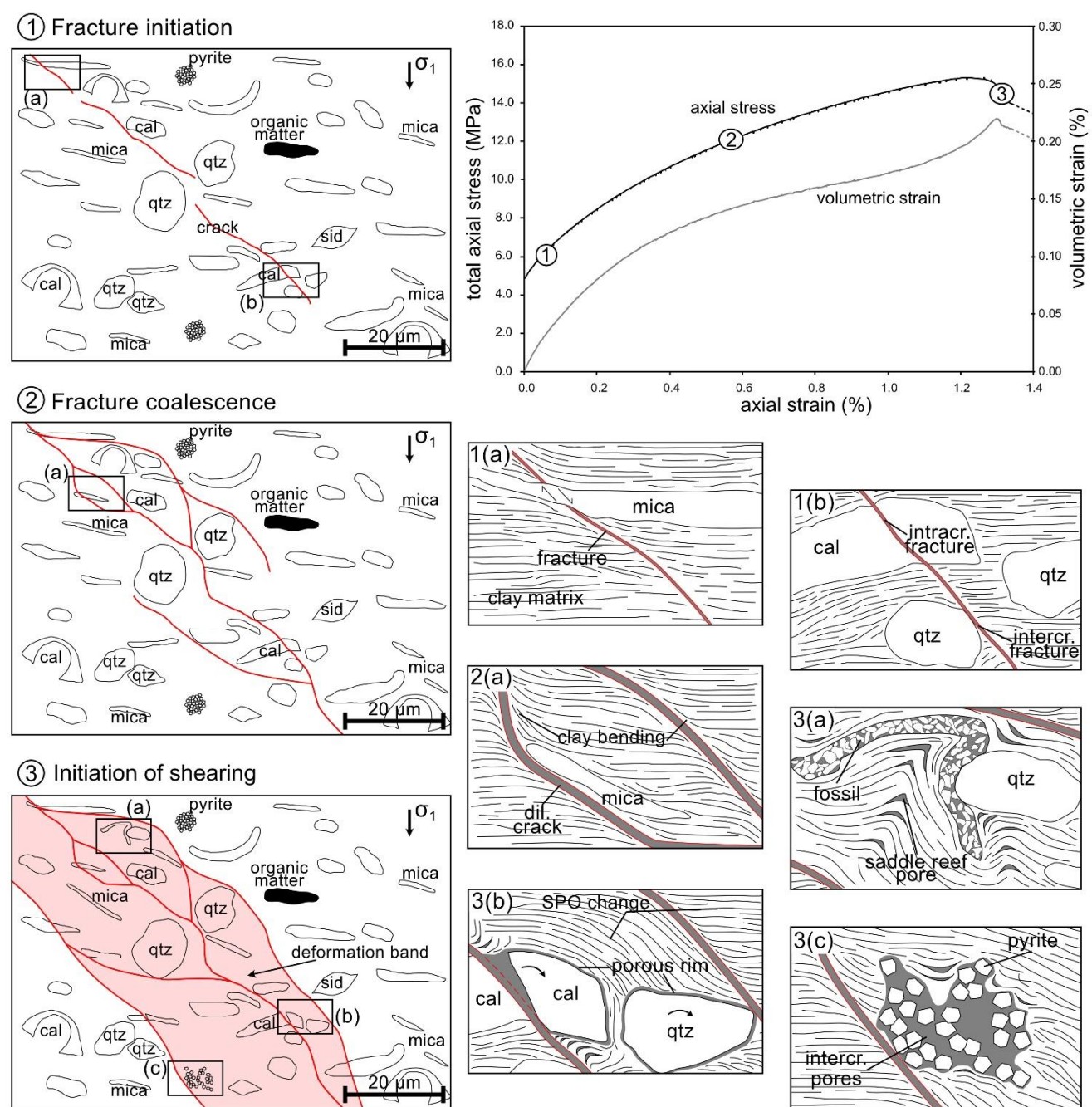

**Figure 9: Deformation model for experimentally deformed OPA in a triaxial test consisting of different stages of damage accumulation: Stage 1 is associated with fracture initiation where fractures form along a preferred oblique orientation. Stage 2 is characterised by micro-fracture coalescence and the formation of anastomosing fracture networks. Here, fractures dilate and clay particles bend towards the fractures. In stage 3, deformation bands show highly deformed microstructures as strained fossils, delaminated clay aggregates, dilatant micro-fractures and a SPO change. Within these bands, porosity is increased by 'saddle-reef**

pores' between clay layers, collapsed fossil shells, porous rims around larger grains and the formation of intracrystalline pores in pyrite aggregates.