# Peer review of "Micromechanisms leading to shear failure of Opalinus Clay in a triaxial test: A high-resolution BIB-SEM study"

_Solid Earth, 2021_

## Author Comment (AC1)

Dear reviewer,

We would like to thank you very much for the thorough review and the critical comments. Your constructive suggestions and corrections strongly improves the quality of our manuscript. In the attached document, we present our changes and corrections to your individual comments.

Kind regards,

Lisa Winhausen and co-authors

SCIENTIFIC COMMENTS

General scientific points that require attention are as follows:

A) The authors often describe their work as addressing the microphysics or micromechanics of shear failure of the OPA material. However, there is no physics or mechanics in the paper at all – it is purely qualitative and conceptual. Rather, the authors address the MICRO-MECHANISMS involved in deformation and failure of the material studied and should really use this term to describe their work (it is a micro-mechanistic study).
→ This is a good comment and we agree on that. However, we feel that also the term 'micro-mechanistic' model implies physical and mathematical equations and laws. Therefore, we used the term 'Micro-deformation' model and replaced all text passages where 'microphysics' or 'micromechanics' is misused.

B) The experimental sample investigated was tested unconsolidated and undrained at 4 MPa confining pressure. There is no discussion as to how these test conditions (hence observed processes) might relate to conditions of interest in the subsurface, such as caprock or repository conditions. Perhaps the effective stress would be comparable (low) due to consolidation effects, but some discussion of this would be welcome – as would the likely pore pressure evolution in the experiment (perhaps based on volumetric strain data).
→ We agree with this statement and included a discussion on the testing conditions and to the application purpose.
*"Many of these processes become effective at long time scales under natural conditions. In tunnelling however, low-permeable rocks such as OPA are subjected to undrained loading conditions during excavation. Thus, the triaxial compression tests have been performed as classical UU tests without prior saturation and consolidation. The pore water pressure development and effective stress inside the sample remain unknown. The test provide the undrained shear strength representative for the nearfield of a tunnel excavated at the consolidation conditions found at the MT-URL(Martin and Lanyon, 2003). Therefore, micro-deformation processes observed in the sample are considered similar to those found in the near-field of a freshly excavated tunnel in Mont Terri. This is further corroborated by the similarities between microstructural EDZ and BDZ (borehole damage zone) observations (Bossart et al., 2002, 2004; Yong et al., 2007; Nussbaum et al., 2011) and microstructures found in this study.".*

C) The description of the experiment does not seem quite right in terms of boundary conditions applied to the sample. The authors seem to be saying that the test was carried out at constant confining pressure and at controlled circumferential strain rate, which is equivalent to controlled radial strain rate. Do the authors mean that axial loading system was servocontrolled to provide a constant radial expansion (or contraction rate)? This seems unlikely to me. Or do they mean that a constant axial strain rate (or stress rate) was imposed and the radial expansion/contraction was simply measured (not controlled)?
→ Our experimental description is correct (compare also Amann et al. 2012). The axial loading is applied in such a way that the sample exhibits a constant circumferential displacement rate. This approach is also described – even tough for unconfined tests – in "ISRM suggested method for the complete stress–strain curve for intact rock in uniaxial compression" (Fairhurstt & Hudson 1999) and

is especially practicable when the axial stress-strain is not monotonically increasing as commonly the case for soft, clay-rich rocks.

D) The terminology used to describe the various cracks, fractures and shear bands at different scales is somewhat confusing to the reader (at least this one!) and not very systematic. I would strongly recommend defining the terminology to be used for these features at an early stage in the manuscript. It would be useful if the authors could define cracks versus microcracks, as well as defining crack character – i.e. whether individual cracks are Mode I (opening) or Mode II (shear) cracks or mixed mode. The term "shear band" should also be defined and appropriately distinguished from a shear crack – at least in a morphological sense. Voids that open between grains due to local dilatation should also be defined as something like dilatational or dilated pores. Of course, the observed features may ALL have been influenced by unloading (removing axial stress and/or confining pressure), so that needs to be mentioned.

→ We agree with this comment and included a paragraph in which we explain the terminology used in our manuscript. We omitted the term 'cracks' and used the synonym 'fracture' instead.

*"Prior to the qualitative description of macro- and microstructures encountered, we define the terminology for our observations and the subsequent discussion. First of all, we use – where possible and not explicitly implied in the section title – prefixes as 'macro' or 'micro' for structures which are visible with the naked eye or under the SEM, i.e. under magnification of at least x100. In general, we subdivide where possible the general term 'fracture' – the surface formed through an initially intact solid material – into 'tensile', 'shear' or 'hybrid fracture' according to their movement along the surface, i.e. normal, parallel or a mixture of both, respectively. Furthermore, we call broader zones of multiple fractures and deformed microstructure 'deformation band', in case for elongated zones with increased porosity 'dilation band', and where shearing is interpreted, we use the term 'shear band' (cf. Du Bernard 2002)."*

E) While it is clear that many of the observed cracks and shear bands are directly associated with the evolution of the macroscopic shear failure process, some features that the authors link to shear failure could potentially be present in the starting material or could have been caused by the application of confining pressure (i.e. to the hydrostatic component of consolidation). Crushed fossils, disrupted pyrite framboids, buckling of phyllosilicates etc all come to mind here. I do not doubt the authors' interpretation necessarily, but it would be desirable to add a few micrographs AT LEAST of an undeformed control sample to prove that the above features are not present in the starting material. Ideally one would like to see a few shots of a hydrostatically loaded microstructure too, if easily available.

→ We understand the reviewers point and argue here that the deformation markers such as micro fractures, crushed fossils etc. are absent in the starting material, i.e. undeformed host rock, as extensively described by Houben et al. 2013,2014. Considering the stress conditions of the over-consolidated host rock from MT-URL with similar stress conditions as used in our study (approximately 4 MPa) we argue that no deformation markers described here are due to isotatic loading as these are well below the past maximum in-situ effective stresses. Furthermore, Winhausen et al. 2021 performed hydraulic testing on shaly OPA at hydrostatic loading conditions of up to 24 MPa effective stress. They performed a microstructural analysis by comparing the unloaded and loaded sample, and found that hydrostatic loading and consolidation does not change the microstructure forming shear or 'damage' indicators as described above. For addressing this point, we added a paragraph to our manuscript and see no need in adding additional micrographs.

*"The deformation markers (Fig. 4-7) are inferred to have formed due to shear failure since these are absent in the microstructure of tectonically undeformed OPA samples from MT-URL (Houben et al. 2013, 2014). Furthermore, Winhausen et al. (2021) analysed the microstructure of a shaly OPA sample, which has been used for hydraulic testing at maximum 24 MPa effective stress, and found no deformation features. Their comparison to a non-loaded twin-sample as well as to other microstructural studies on OPA indicate no structural damages due to isostatic loading at this effective stress (Winhausen et al. 2021).*

F) Last but not least, some brief discussion is needed as to how fluid flow from consolidating regions of the sample to dilating regions of the sample may have facilitated shear localization (as opposed to dilatancy hardening), as this could be a key component in controlling the failure behavior and strength. Such aspects would be essential for any quantitative microphysical modelling in future, and notably for understanding strain rate effects.

→ We agree with the reviewer that the dilating regions play an important role, but see here the opposite effect: As the whole test was performed in less than 40 min, consolidation processes in this low-permeable rock do hardly play a role as the conditions within the sample can be considered undrained. However, we agree that the local dilatancy creates more pore space and, therefore, a reduction in pore water pressure and a local increase in effective strength in the sheared zones. We see here two competing mechanisms to act: The facilitated shearing (shear softening) due to the loss of cohesional bonds and the effective strength increase (shear hardening) within the shear zone. Based on the only sample analysed in this study, we argue that shear softening is the prevailing process but unequivocally strain rate effects play a great role (cf. Giger et al. 2018) when testing at consolidated boundary conditions. We will not extent the discussion in the manuscript but empathize this aspect in our ongoing studies under consolidated testing conditions. Nevertheless, we included a brief statement in this discussion to our text:

*"The local increase of porosity in the deformation band leads to a local reduction in pore water pressure under undrained conditions and a related local increase in effective stress in the shear zones. The increase in effective stress involves a shear hardening effect, which represents the competing effect of shear softening due to the loss of cohesional bonds. Here, further research is required to investigate the dependency of permeability and porosity evolution with increasing amount of shear stress and the effect of undrained and drained conditions within the sample.*

Detailed scientific comments/questions

1) Title: This seems unnecessarily long and not especially informative to me. Would something like "High resolution BIB-SEM study of micromechanisms leading to shear failure of Opalinus Clay in a triaxial test" not be more accurate and sufficient??

→ We change the title to be very similar to the suggested one

*"Micromechanisms leading to shear failure of Opalinus Clay in a triaxial test: A high-resolution BIB-SEM study"*

2) Abstract: Best clarify that failure occurred by shear failure and mention orientation of the shear fracture/band network (line 21).

→ Both point were added to the abstract

*"Observations on the cm- to μm-scale showed that the samples exhibited shear failure and that deformation localised by forming a network of μm-thick fractures which are oriented with angles of 50° with respect to horizontal."*

3) Last statement about LIMITED similarity with natural fault zone microstructure and inferred role of pressure solution in nature is not very well supported in the main text. Perhaps add some additional evidence in the main text?

→ We added some additional similarities to the main text and the new following paragraph is discussion the existence of processes occurring in experimental vs. natural conditions.

*"Some structures found in this study, such as the anastomosing fracture network, strained fossil shells and bending of phylosillicates resemble those found in highly strained OPA (Laurich et al., 2017; Orellana et al., 2018)"; see B)*

4) Intro: I advise using the term "radioactive waste" rather than "nuclear waste". Nuclear suggests high level waste risk which is often not the case. Atomic nuclei are also hardly waste.

→ We replaced the term accordingly.

5) Page 2, line 48. Microstructure is perhaps the wrong term here. Mineral composition would be accurate. Line 52 – what measure of visible pore size shows a power law distribution??? Lines 61-62 – compressive loading of samples in what orientations with respect to bedding? And what is meant by plastic flow – please define as it has many meanings.

→ We replaced the text according to the suggestions and added the orientation of the samples tested in the cited reference

6) Page 3, Line 83: Is "slickensides" the correct term here for these very fine striations in the mirror slip surface? Are they optically visible or only in SEM and sub-optical? Line 88 - do you mean smectite interlayer water here? Please clarify.

→ We used the terminology as given by Jordan and Nüesch, 1991. Slickenslides is not referring to the striations but the shiny and mirror-like surfaces (on which striations might be visible). This is well explained in the text. Jordan and Nüesch 1991 observed these both optically and in the SEM. We added this missing information to the text. Concerning the water inter layers, the cited authors (Jordan & Nüesch 1991) mean the absorbed water between clay two layers but do not specify the clay mineralogy they are referring to. We changed the sentence slightly.

*"These shiny and mirror-like surfaces (i.e. slickenslides), optically and microscopically visible, consist of denser packed clay material."; "[…] as 'sliding surfaces' on water interlayers of clay minerals and […]"*

7) Page 4, Line 124: Mixed layer silicates?? Do you mean mixed layer smectites and mectite-illite??? Last line: what is the pore fluid? A brine with what composition?

→ We specified in the text which clay mineralogy is included; detailed subdivision of different clay minerals is not provided by the authors (klinkenberg et al. 2009) due to missing Rietveld analysis. The sample was not re-saturated by an artificial brine. Full saturation was demonstrated by porosity and water content measurements as explained in Amann et al. 2012, which is referenced in the text.

*"[…] clay minerals (50-66%) composed of 2:1 layer with illite, muscovite, smectite, and illite-smectite mixed layer minerals are (20-30%), […] "*

8) When first describing fracture/crack/bedding/loading orientation, please define what you mean by orientation or inclination in terms of an angle (say theta) defined in a small diagram. Present usage is confusing, perhaps due to language. It is important to be consistent in the use of this terminology throughout the paper. Confusing at present.

→ According to this and the third reviewers suggestions, we included a sketch (included in Fig. 3) for the definition of the inclination angle. Furthermore, the directions of bedding and loading were included in the figures, where still missing. We added a sentence to the text referring to the diagram added to fig.3

*"In the following, all angles are defined as deviations from the horizontal, i.e. load direction of σ3 (see diagram in Fig. 3)."*

9) Section 2.2. Please clarify boundary conditions imposed in the experiment – see point B above.

→ We comment on the selected boundary conditions in the discussion. Please, see point B)

10) Section 2.3, first line. Mechanically stabilized with epoxy ?

→ The sentence has been changed accordingly.

11) Section 3.1, last 3 lines. Specify branching fractures as the green ones if that is what is meant (all look branching to me). Fragments are only CRUDELY lens-shaped – best say this perhaps. Please clarify what is meant by "relay fractures".

→ We changed the text accordingly and inserted a small zoom-in illustration showing the relay structures observed on the macroscale to Fig. 3.

12) Section 3.2, first paragraph: How long were the samples stored before study ?? Please indicate in the material description. What reacted to produce gypsum? How do you know this occurred during sample storage?? Could storage have had any other effects that could corrupt the microstructure?

→ We added the missing information to the text (sample storage of five years) and explained why gypsum formation is likely to happen after core extraction (when exposed to air). Storage can lead to swelling or shrinking of OPA due to the changing stress and de-/saturation conditions. These effects are mentioned in our discussion.

*"[…] by chemical reactions during sample storage before or after the experiment. When exposed to air, pyrite will oxidise creating sulphate, and the presence of Ca-ions from calcite (and dolomite) will lead to the formation of gypsum. Timing of this process could not be determined and, therefore, these regions were excluded from microscale analysis."*

13)Page 6, Line 183: Microcracks were often oblique to bedding…. Which set is meant here?
→ We changed the text accordingly.
*"Micro-fractures, which belong to fracture set 1), were often […]"*

14) At several points in the description of the microstructure (Results – Figs 4 and 5), reference is made to increased porosity and grain-matrix separation at specific sites around clasts and clast-like/sigmoidal micro-lithons. These sites seem to correspond to interfaces under local extension – i.e. at roughly 45 degrees to the overall (macroscopic)shear failure plane orientation. This should be described as such, if indeed a widespread feature, making reference to previous work that has observed and constructed models for this type of feature (e.g. Den Hartog & Spiers 2014; Haines et al., 2013). These featureshave key mechanical significance in defining the onset of mechanical weakening due to a decrease in dilation angle (see papers by Den Hartog et al).
→ We understand what the reviewer is referring to but do not entirely agree with his/her observations. In fact, in Fig. 4 can be seen that grain boundary-matrix slitting appears in all directions (calcite fossil in centre for instance parallel to "shear plane" as of larger, rounded quartz grain in lower third shows a rim with orientations from parallel to 90° to shear band direction. Similar observations can be made in Fig. 5a), where quartz grains in the upper right corner are completely surrounded by porous rims. We can see the 45° degree trend in one calcite grain (upper right, label "cal"), but again the siderite grain (lower right, label "sid") shows a grain-matrix separation which is parallel to the macro shear zone. We also want to state here that we present microstructures of an incipient "shear zone", the MS of the sample does not show fully developed shear bands or gouge.

15) Page 8, line 235: "Plastic reorientation of clay aggregates". What exactly is meant here? Why is this plastic not frictional intergranular sliding? "Ductile" deformation is inferred from reorientation to form an SPO, I guess? The term "plastic" is used elsewhere in the ms to refer to intracrystalline plasticity of the phyllosilicate grains> perhaps make more consistent? Line 236 – "grain boundary sliding". This term is generally used for a high T process involving grain boundary dislocations. Better use "intergranular sliding" here as it is likely a frictional or fluid-lubricated sliding process. Lines 239-242 – I am surprised that no reference is made here to the top quality work on deformation of claystones such as COX by the Paris Est (Ecole des Pont) Team (recent papers by Philippe Braun, Pierre Delage and co-workers). Relevant work should be added.
→ In line 235, we deleted "plastic" which was not appropriate in this sentence. Throughout the manuscript, we distinguish between brittle (breakage/fracture with prominent surfaces) and ductile deformation (bending and reorientation). We changed all parts according to make the text consistent. We replaced "grain boundary sliding" by "intergranular sliding". Indeed, we know the work of Braun, Delage et al. However, they performed hydro-mechanical experiments on (mostly) isostatically-loaded samples to derive the material's poroelastic properties and dependencies on temperature. Up to our knowledge, they performed no triaxial deformation experiments in which they analysed the (micro)structural deformation and we see therefore no valuable contribution to our manuscript.

16) Page 9, Line 260: It is assumed that unloading elasticity is equal throughout the sample. Perhaps make this "roughly equal" as the elastic stiffness of the porous zones will certainly be much lower than the undeformed matrix. Line 268-269 – reference is made here to the amounts of elastic versus inelastic deformation in the sample. This could be evaluated from the elastic part of the unloading curve, perhaps. Not done? Could this be added to add a littlle more quantitative information?

→ We changed the sentence according to your suggestion. Unfortunately, we have no data from the unloading part of the experiment, hence, we cannot evaluate this quantitatively.

17) Section 4.3 - Micro-mechanical model. This is a misnomer, I feel, as the model presented is a conceptual model containing no real micromechanics – a term that implies a quantitative approach. This header is perhaps best modified to "4.3 Micro-mechanistic model" - which is probably what is meant by the authors. Perhaps it would be wise to add to this section the caveat that the conceptual model developed is based on a low P experiment with no pre-consolidation, and to clarify the imposed boundary conditions (constant P, controlled axial or radial? strain rate etc etc). It has to be recognized that different microstructures may develop under different conditions, or it should be argued why this would not be the case.

→ We agree and inserted the missing information to this paragraph. We also changed the section title and replaced the phrasing in the text accordingly; see also point A)

*"For the unconsolidated, undrained deformation of OPA under differential compressive load at a total confinement of 4 MPa, we propose the following deformation mechanisms based on our microstructural observations […] "*

18) Fig 9 vs. Fig 1. This depicts the micromechanistic model proposed. However, the stress strain-curves used to illustrate the various stages of evolution looks rather different from the original experimental data depicted in Fig.1. Is the stress axis perhaps the total axial stress as opposed to the differential stress plotted in Fig. 1?? Please clarify as the stress levels mentioned seem not to match up as is. It would be useful to clarify the orientation of Sigma-1 in the microstructural sketches also. In Fig. 1, how was the stress level labelled "crack initiation" established? This should be mentioned somewhere?

→ Yes, the plot in Fig9 shows the total axial stress instead of the differential stress, we added "total" to the axis title; apart from that, the plot shows the same data as in Fig.1. We inserted the orientation of sigma1 to the sketch. "crack initiation" or "fracture initiation" as used in this manuscript has been adapted from Amann et al. 2011. They determined this threshold as the deviation from the linear-elastic behaviour of the volumetric strain curve. We added the explanation to the text.

*"Amann et al. 2012 showed that fracture initiation, the point where the volumetric strain curve deviated from linearity ("crack initiation" in Amann et al. 2012), occurs long before stress peak at approximately 2 MPa differential stress independent from confining stress."*

19) Section 5 - Implications and conclusions. As indicated above and in my general scientific comments, this section would benefit from some brief consideration of the extent to which the observed processes and proposed mechanistic model can be expected to apply under shallow upper crustal (geo-storage system) conditions. The authors do state that more work is needed on consolidated samples and on naturally sheared/faulted samples to test the broader applicability of the model. However, in future experiments, would not higher P-T conditions and deformation/loading rate not be important too, bearing in mind issues like smectite hydration state changes and internal fluid pressure equilibration, for example?

→ We agree with this point and included a paragraph to section5

*"Our high-resolution study gives insights into the macro- and microscale structures with some similarities to excavation or naturally induced structures. However, OPA - deformed up to peak stress - also shows structural differences to naturally deformed OPA, which shows drastically reduced porosity. Strain amount and rate in natural fault systems, as well as elevated temperatures or clay-mineral hydration processes may influence the deformation mechanisms and structures. On the other hand, natural veins as an indication for paleo-fluid flow in the early faulting stage (Clauer et al., 2017) support localised dilation as similarly found in this study. Future experiments covering a broader range of*

*boundary conditions on consolidated samples are required to analyse the deformation processes under shallow upper crustal conditions, and to formulate an effective stress dependent deformation model.*

20) Still Section 5: In Line 353 of page 12, how does the present study support implementing damage into constitutive modelling in the references mentioned? Do the authors mean "support" here or "provide input to constrain" such models, perhaps?
→ We changed the sentence according to your suggestion.
*"The results of this study support more recent efforts […] to provide input to constrain damage in constitutive models […] "*

21) Figures 2-7: It would be useful if the orientation of the principal compression direction is indicated in all key micrographs, either directly in the micrographs or appropriately via the captions. Angles defining fracture and shear band orientations should be marked on key micrographs also, so that the intended meaning of terms like inclination (to what? Horizontal? Vertical? Bedding? ) is clear. What is OM in Fig. 5a?
→ We inserted a sketch indicating the orientation of bedding and macro-shear band orientation in each key micrograph with according colour labels as presented in Fig.3, and added a description of such in the figure captions. We replaced OM by "organic matter".

TECHNICAL ISSUES (language, typographics etc)
Overall the paper is well written and in good English. Nonetheless a few small improvements can be made as follows:
i) Fault gouge spelling: Gouge NOT Gauge → corrected throughout the text
ii) "Associated with" is the correct usage, not "associated to" → corrected accordingly
iii) "Indicated "should not be used to mean "correspond to". Use the latter. → exchanged where necessary
iv) A typo: Saddle-reef pores. Better define the term also as it is from structural geology and is not familiar in rock mechanics. → corrected and definition inserted
v) "Acting on"… not "acting to..". → corrected
vi) "growed" >>>> grew → corrected
vii) Page 11 – reformulate first sentence. English awkward/unclear. → sentence reformulated
Finally, I would recommend a thorough spelling/grammar check at the end of the road. I wish the authors good luck in revising the paper and hope that my comments are helpful and not too burdensome. Please note that I have elected not to do a second review, not out of disinterest but from "review overload".

---

## Author Comment (AC3)

Dear Ben Laurich,

We would like to thank you very much for the thorough review and your suggested changes. Your constructive input strongly improves the quality of our manuscript. In the attached document, we present our changes and corrections to your individual comments. Your additional changes in the attached word document "se-2021-39-BL" are much appreciated; our comments and changes are listed below as well.

Kind regards,

Lisa Winhausen and co-authors
* * *
Main critical points:

1. Although dilation bands in OPA are shown for the first time, the finding of high permeability in the early strain evolution is not a novel finding. Veins along fractures in the so-called Main Fault of the Mont Terri Rock Laboratory have long been reported as indicators for paleo-fluid flow by numerous authors (see attached word file). This should be addressed.
→ We added this point to the text by including your suggested statement in 4.2.
*"At the same time, a local increase of porosity may imply an increased permeability. This is striking and important to note since µm-thin shear zones from the 'Main Fault' in OPA of the MT-URL usually present a decreased porosity (Laurich et al., 2017) and deformed OPA in direct-shear experiments shows a reduction in permeability (Bakker and Bresser, 2020). However, fracture-sealing veins (mostly calcite) in the Main Fault also indicate a paleo-fluid flow. Isotopic studies suggest that these veins formed contemporaneously with the initial fault activation, which supports or results in localised dilation in the early stage of shear development (Clauer et al., 2017)."*

2. The use of the term "ductile": The stress-strain curve shows bulk brittle behaviour, yet there are two indicators for an uncritical strain evolution in parts of the sample: (1) bend and delaminated clay grains as well as (2) finely distributed brittle shear within deformation bands / cataclastic flow. These three scale-depended phenomenon (bulk, grain-scale and deformation band-scale) could be better distinguished and elaborated more. For each occasion of the term "ductile": To what scale does this refer to? Optional it could also be addressed: At what condition (P, T, strain rate) might a rather stiff mica grain bend? At what condition does a calcite grain break? At what condition does sub-critical fracture grow happen instead of seismical fracturing?
→ In section 4.1 we inserted the scales where processes are described. Furthermore, we inserted scale dependencies in the paragraph where we describe the deformation model.
*"Clay bending and intracrystalline fractures correspond to a coexistence of brittle and ductile deformation processes on the grain scale due to the contrasting mineral rheology." "These deformation bands are characterised by both brittle and ductile deformation in form of matrix cataclasis and reorientation of mineral fabric on the µm to tens of µm scale, even though on a bulk scale the sample deforms in a brittle manner."*

3. Terminology: There is a little inconsistency in the use of deformation band, deformed zone, strain zone, shear zone, fracture and crack. I gave some suggestions on how to improve that in the world file. From du Bernard (2002), there is the term "dilation band", too.
→ As also suggested by the other reviewers, we inserted a paragraph on our used terminology for describing the microstructures. We replaced 'deformation zone' by 'deformation band' and adapted, where possible, from Du Bernard 2002.

4. A statement on subsample positioning is missing. Might there be a spatial relation on strain localization and strain rate to the subsample position? The crack likely nucleated at the sample center, where there is a high abundancy of fractures, that distribute the bulk strain while at the sample edge there is only one fracture. Has brittle deformation started once that first single fracture reached the samples' edge to create one sample through-going, cohesionless discontinuity? Might a shear zone at the samples' edge represents a more brittle deformation, while a shear zone from the samples' center might have preserved more of the prior-to-failure, uncritical fracture growth microstructure? A subsample off-center in the middle, such as in Fig. 6 providing insight in the most less and slowly strained shear zones?
→ We agree that there is a chance for different localized strain distribution. However, from our results we do not wish to relate the spatial distribution and the deformation mechanisms/microstructures due to missing evidence for this assumption. Presented deformation marker and MS has been found at several locations within the sample. Furthermore, many of the high-aperture macro fractures, i.e. where higher shear strain localisation is expected (distributed both close to the top and in the centre), were inferred to contain artefacts from sample preparation and are therefore excluded from analysis. We included a statement on this in the last section of the discussion 4.2:
*"We note here that the microstructure in vicinity of the larger macro fractures, i.e. close to the "main" fractures, where potentially most of the shear strain is accommodated, host microstructural artefacts from sample preparation (cf. section 3.2) and we hypothesise that with ongoing deformation shear strain will localise along a macroscopic shear band cross-cutting the sample with a reduction in porosity and grain sizes."*

5. The title sould be shortend, see attached word file
   • We changed the title according to your and the first reviewers suggestions.

For a point-to-point discussion, we inserted the major comments (typos, spelling mistakes etc. have been taken over as suggested) from the word file:

• reference to Busch et al. (2017)
   → We did not only compare to MIP measurements but to most of the methods used for porosity measurements by the authors (Busch et al. 2017), i.e. water content porosity, He-pyn., neutron scattering. We added the missing techniques to the text.
   *" […] OPA of 15.3 % (Houben et al., 2013, 2014). This value is slightly lower than petrophysical porosities measured by various methods such as water content porosity, Helium-pycnometry and neutron scattering techniques (Busch et al., 2017)."*
• nomenclature on fractures/cracks/joints/shear zones, etc.; apature/width; general remark point 3)

→ We inserted a paragraph on the use of terminology for the manuscript (first paragraph in section Results). We decided to use the term fracture instead of crack, as many times it is not clear if fracture has a shear component or not. For describing fractures we used the term aperture, for shear /deformation zones we used width as suggested.

- This is interesting to observe in this rather rapid laboratory test. I looked for these "non-fractured" microstructure but couldn't find them in in-situ OPA. Maybe this is as only slickensides expose themselves easily. Finding this non-fractured but strained volume is like the famous needle in the haystack. I'd suggest to stress this finding.
Plus, to me "ductile" behaviour of OPA was mostly thought of as finely distributed brittle shear (cf. scaly clay). Here, instead, it seems to be a grain-scale mechanism that differs from brittle. What is it though? Passive grain rotation as in the S/C bands of gouge? If so, these def. structures must be more dilatant in their early evolution to allow for grain rotation. See also main critical point #2.
→ We stressed the scale-dependent behaviour by including the scale for the different mechanisms at multiple locations in the text (and also the conclusions). We also included the meaning of 'ductile' behaviour where used in the text, e.g.:
*"Some of these deformation structures showed less dilatancy, mostly ductile deformation and a more pronounced shear component on the grain scale (Fig. 5(c)). Here, the SPO of non-clay minerals and clay aggregates was oblique with respect to the bedding and curved according to the shear sense of the micro-fracture."*

- Such a drastic rotation can be two-fold: either, as in your fig 6, by the rotation of individual grains. Or, as within scaly clay, by the rotation of rigid microlithons. In the latter case, the MS is unchanged but rotated as a whole. In your case of fig 6c, I'd suggest to rewrite: "….the original SPO was broken up and elongated grains reoriented to a parallel alignment with shear movment."
→ We changed the text based on your suggestion.

- Where in Figure 6c is your "deformation zone"? I'd say the entire 10μm is a "shear zone", including the broken-up SPO grains.
→ We inserted the orientation of the shear zone to a micrograph (e) which applies for all the micrographs in this figure (mentioned in caption). From this, it can be seen that the SPO has rotated to be (sub-)parallel to the shear zone.

- So deformation bands are precursors to shear zones? There are dilatation bands first (du Bernard, 2002) and they evolve further?
→ We define our terminology in a newly added paragraph which hopefully clarifies the terminology used and avoids confusion. But yes, we interpret that shear bands form from deformation/dilation bands. One has to keep in mind that we do present a microstructure where "shearing", in a sense of localised shear strain along a distinct plane/broader zone, has just initiated; therefore, we do not call the "bands of deformation" rather "incipient shear bands".

- According to the paragraphs above, ductile indicators are: (1) bend mica and (2) delamination of clays, (3) maybe passive, sub-critical rotation of grains (see my comment above). However, (2) and (3) could be justified as brittle processes, too (see your next sentence). The problem here really is the lack of a good, comprehensive definition for "ductile" that is valid on all scales. Anyhow, at least the qz and ca grains do not seem to be deformed in a ductile manner. So I suggest to reword this sentence.

→ We agree on that and changed the sentence to stick to our definitions of brittle or ductile deformation, and to be consistent throughout the manuscript.

*"[…], the deformation of both the clay-rich matrix and larger quartz, calcite and mica grains as well as pyrite aggregates is governed by brittle or ductile processes. On the one hand, grain scale clay splitting, grain abrasion and intra- and intergranular fracturing points to cataclasis localised in the deformation band. On the other hand, reorientation of clay aggregates and bending of elongated grains indicate structural reworking on grain scale to enable granular sliding."*

- "… the ratio of porosity within and outside the deformation bands remains constant." I do not see any justification for this assumption. The deformed zones are cohesionless, the undeformed matrix has (little) calcite cementation. Maybe argument by using references that quantify the shaly facies' cohesion. Bear in mind that Houben et al. (see above) argue contradictory for unchanged, in-situ resembling MS & porosity no matter what drying method used.
The idea of stored elastic energy from diagenetic bonds by Corkum et al. explicitly state that a break-up of the bonds is necessary, correct?
→ We agree that the equality of elasticities is an assumption and cannot be proven by our data. We changed the sentence accordingly by rephrasing it for a more hypothetical view. Nonetheless, we explain many aspects which bring strong evidence that the porosity in the deformation bands is increased in the experiment.
*"If we assume that during unloading the compressibility is roughly equal in and outside deformation bands and, then the ratio of porosity within and outside the deformation bands remains constant. Keeping all the above in mind, we infer that the deformation bands formed a local increase of porosity during the experiment."*

- Circular argumentation. Elastic strains are recoverable by definition.  If what is meant is the incomplete recovery from the strain before elastic yield limit (deviation from linearity in Fig.1 / hysteresis) than it cannot be inferred from your observations, so maybe add a reference.
There is even an indication that shear zones from differing OPA facies have a differing tendency for strain recovery, i.e. hysteresis, in that pre-elastic-yield region:
Laurich, B., Kaufhold, A., Hertzsch, J.-M., Gräsle, W., 2018. DB Experiment: Rock mechnical lab testing on samples from the BDB-1 borehole, Technial Note Mont Terri Consortium. Hannover, Germany.
→ We agree that our sentence was meaningless and inappropriate for the discussion; hence, we deleted it and changed the next sentence to fit the logical statement intended.
*"Elastic deformation took place by deformation of the solid component and of pores – this was inferred to be reversed by unloading the sample."*

- Why non-localised? Would that mean finely and manifold distributed crack nucleation?
You write above that the orientation and distribution of cracks remains unknown. However, triax-samples often fail somewhat in the middle of the sample length. In almost any case (as with this sample), there is a locally distinct failure. I reckon that this plastic failure is preceded by a local accumulation of elastic strains, too.
Plus, Fig. 3 shows a higher fracture abundance in the center, while only one fracture reaches the sample edge. This indicates sample internal failure / strain initiation. With cohesion overcome by the first (and only) fracture reaching the sample edge, displacement sets in rapidly (Fig 1) and

doesn't form new fractures but offset only on the existing fracture (argumentation for a dominance of brittle processes close to and after bulk sample failure)

See also your own argumentation below.

→ The crack/fracture nucleation, i.e. at an early stage of deformation, is interpreted to happen in a non-localised manner in the sample with a denser accumulation in the center of the sample. This has been demonstrated for brittle deformation rocks (e.g. Kranz 1983 and references therein) measured by micro-acoustic emission tests and was observed on our sample by multiple micro-fractures of set 1(green) with tensile splitting. We do not state that the fracture coalescence and later shear strain development is non-localised, we rather say the opposite. You state that the failure is along a larger fracture in the middle of the sample – and this is true – but in this sentence, we do not refer to the failure of the sample. We added the reference.

*" […], and it is interpreted as a non-localised process within the sample as reported in other studies (Kranz 1983 and references therein)."*

In section 4.3, we changed the sentence slightly in agreement with the observations and our interpretation

*"These fractures form in the whole sample with preferred orientations crossing the bedding and with ongoing deformation they concentrate with a higher density in the centre."*

- Only an optional suggestion: I don't think there is any evidence-micrograph for this in any study. But if so, I suggest to cite it here. Why is this important to me? I reckon that these uncritical crack propagations are controlled by several environmental factors, foremost by strain rate. Insight here could justify the comparability of lab and in-situ crack formation.

  → We missed to insert the reference to Fig. 5C, where indeed ductile deformation, in close to fracture tip regions and at the boundary between deformation zone and wall rock were observed. We agree that strain rate plays an important role for the deformation structures and related mechanisms. We added a paragraph that to the discussion 4.2 second last parargraph and conclusion.

- Can you provide an estimate on the displacement, too? How many μm? For a shear zone from the MF in MTURL, I found a tool mark of about 2 cm length.

  → As stated in the manuscript, we could not find correlation indicators, hence no estimate on the displacement was possible.

- Not necessarily: The def. band can change from dilation, leaving space for first particle reorientation, to compaction band, wherein μm-thin shear zones with slip-aligned particles and reduced porosity can also cause the bulk compaction by a reduction in the bands' width.

  → This is a possible hypothesis; however, in our study we could not find any indication for the closure of the deformation band. Hence, we did not include this into our discussion.

- Why is a deformation band preservation favourable over the development of a single, through-going shear zone such as the Y-shears? The later would be energetically more favourable, as frictional forces would be reduced. And if I get it correctly, that is what you observe in Fig. 5c and Fig. 6 a, c, d: Movement on a distinct deformation band boundary. Why is this important to me? Does this relate to the formation of in-situ scaly clay? There it is one potential evolution that the internal structure of a band (in the scaly clay case an S/C band) form passively in between a (growing) relay of two bounding μm-thin shear zones that accommodate most of the displacement.

Here, it seems to be the other way around: dilation band first, then further localization at a boundary. Could this effect of grinding down microlithons during the preservation of a deformation band have a lower size limit? If so, will it eventually form gouge or a single, through-going fracture or, as you seem to suggest, maintain a smallest microlithon size? (cf. Laurich et al, 2017, 2018).

→ We agree that the full analysis of the formation and development of the deformation bands is still poorly understood and that more research (more tests and MS analyses) is required here. However, we present our interpretations based on this only test and will not hypothesise beyond the scope of this study in much more detail regarding their deformation mechanisms as analysed by Laurich et al. Again here, one has to keep in mind that the BC used for this test are somewhat different from the in situ conditions for naturally deformation OPA. We discuss why naturally deformed OPA is different from our observations at several parts in our manuscript and name additional effects that might play a role.

---

## Author Comment (AC4)

Dear reviewer

We would like to thank you very much for the thorough review and your suggested changes. Your input strongly improves the quality of our manuscript. In the attached document, we present our changes and corrections to your individual comments.

Kind regards,

Lisa Winhausen and co-authors

Main comments:

- The authors should consider the role that mechanical anisotropy has on their results. It is mentioned briefly in the introduction, but then very little attention is given to this thereafter. I feel that some of the observations, for example the orientation of tensile cracks being oblique to the loading direction, could be explained by the mechanical anisotropy present within the material. I have suggested some studies which the authors may find useful on the subject.
  → We agree with this point. Therefore, we added a paragraph on the influence of anisotropy on the fracture propagation/deformation in shales by also including your suggested references in the introductory part. We also discuss the mechanical anisotropy in the discussion 4.1 + 4.2 while suggesting that more work is needed on different sample configurations (load direction to bedding orientation) to constrain dependencies.
  1. Introduction:
  *"Additionally, Ibanez and Kronenberg (1993) distinguished deformation modes based on the loading direction with respect to bedding. Individual and conjugate fractures with orientations between 33° to 73° towards horizontal were found in samples loaded normal to bedding, whereas samples loaded parallel to bedding are prone to develop macroscopic kink bands with orientation of 47° to 75° and material rotation. Shear zones with orientations of 56° to 46° including enchelon fractures are typically observed for sample where maximum load is oriented by 45° degrees to bedding. The influence of anisotropy on fracture mode and propagation has also been studied in fracture toughness experiments on shales and shows that the direction of fracture growth is dependent on i) the orientation of anisotropy, ii) the extend of anisotropy and ii) the loading mode (Forbes Inskip et al., 2018; Nejati et al., 2020)."*
  4.1 Discussion:
  *"Fracture kinking or deflection is a common process in anisotropic rocks and is due to the competing influences of applied load direction and the direction of anisotropy plane (Forbes Inskip et al., 2018; Nejati et al., 2020). Our observations suggest that both factors play a role for fracture orientation and propagation since, on the grain scale, fractures cross the bedding and form along bedding (Fig.4a, Fig. 5). However, more work on samples with various loading configurations is required to quantify the influence of mechanical anisotropy on the fracture orientation."*
  4.2 Discussion:
  "Tensile microfractures form as obliquely – instead of vertically – oriented fractures along grain boundaries, which is inferred to be a result of loading direction and the orientation of anisotropy plane."
  *"[…].These anastomosing fracture networks also show the influence of mechanical anisotropy as dilatant fractures form along the bedding plane as opposed to the general orientation of the fracture network (Fig.6(a))."*

- There is an inconsistency when describing the orientation of structures or processes, where the authors use the sample orientation, bedding orientation and loading orientation throughout. I found this confusing when reading the MS and I would recommend that the authors use the bedding orientation only. There is one point (which I highlight below) where it is also appropriate

to refer to structures/processes relative to the loading direction as well, but apart from that I would just stick to bedding orientation.

→ We understand the confusion and changed the orientations in the manuscript to fulfill our definition of angles measured. We measure all angles from the horizontal and inserted a sketch in Fig.3 for clarification. We decided to measure the angles from horizontal as this is – to our understanding – more convenient since the bedding is inclined as well and one would always have to calculate what orientation this would be towards horizontal (or vertical, i.e. principle stress directions). Additionally, we inserted for all micrographs the orientation for both fracture sets measured on the macro scale to better relate the orientations on all scales described. We hope that you will agree with this.

- At the end of the MS the authors run the risk of undermining their own great work by suggesting that their results do not compare well with naturally deformed samples of OPA. This may be the case, but then you need to stress why your contribution is still valuable, and give more details on why you think there are differences, and what can be done in any further work. The authors give a vague answer to this, but I would suggest that they end on a stronger note.

→ We completely agree with this point and changed parts of text by discussing the differences but also presenting some more similarities (Last third of 4.2). Furthermore, we included a comparison to artificially induced (excavation induced) fractures from the EDZ and BDZ (last two paragraphs in 4.2). We also included some more suggestions on what to improve in future experiments. (central paragraph in conclusions) and incorporated these points to the conclusion.

4.2 Discussion:

"[…]. Some structures found in this study, such as the anastomosing fracture network, strained fossil shells and bending of phylosillicates resemble those found in highly strained OPA (Laurich et al., 2017; Orellana et al., 2018) […]"

"[…]. This is striking and important to note since µm-thin shear zones from the 'Main Fault' in OPA at the MT-URL usually present a decreased porosity (Laurich et al., 2017) and deformed OPA in direct-shear experiments shows a reduction in permeability (Bakker and Bresser, 2020). However, fracture-sealing veins (mostly calcite) in the 'Main Fault' also indicate a paleo-fluid flow. Isotopic studies suggest that these veins formed contemporaneously with the initial fault activation, which supports or results in localised dilation in the early stage of shear development (Clauer et al., 2017)."

*"Many of these processes become effective at long time scales under natural conditions. In tunnelling however, low-permeable rocks such as OPA are subjected to undrained loading conditions during excavation. Thus, the triaxial compression tests have been performed as classical UU tests without prior saturation and consolidation. The pore water pressure development and effective stress inside the sample remain unknown. The test provide the undrained shear strength representative for the nearfield of a tunnel excavated at the consolidation conditions found at the MT-URL. Therefore, micro-deformation processes observed in the sample are considered similar to those found in the near-field of a freshly excavated tunnel in Mont Terri. This is further corroborated by the similarities between microstructural EDZ and BDZ (borehole damage zone) observations (Bossart et al., 2002, 2004; Yong et al., 2007; Nussbaum et al., 2011) and microstructures found in this study.*

*These studies show that purely extensional fractures coexist with shear fractures in the EDZ/BDZ. The latter – compared to tectonic fractures – are characterised by poorly developed striations but indicate re-oriented clay particles (Nussbaum et al., 2011). Furthermore, BDZ structures in a resin-impregnated over-core from the shaly facies of OPA at MT-URL showed branching fracture patterns on a mm-scale similar to the structures observed in our study (Kupferschmid et al., 2015)."*

Specific and Technical Comments

Line 18 – Why test at this orientation? This is also a general question, not just specific to this line.

→ This contribution aims at giving a first insight to the grain-scale deformation mechanisms of OPA from a single test, which has been performed in the past. The sample originated from the study Amann et al. 2012, which have tested a series of S-samples (bedding normal to sigma 1) under various confining stresses. We fully agree that this is only one end-member and more tests are required to study the influence of anisotropy on the deformation behaviour and processes. This is stated in the text twice.

Line 28 – Could you briefly state what the major effect on permeability evolution is in the abstract?
→ we added this information to the sentence:
" […], with an inferred major effect on permeability by an increase in hydraulic conductivity within the deformation band."

Line 51 – Are the bedding parallel cracks likely to exist at depth? Or are they only present on exhumed material that has therefore been unloaded? This may be pedantic but the way this reads at the moment suggests that these exist at depth, like the pore space in the fossils, pyrite or clay matrix (which all presumably exist at depth).
→ We included the prefix 'SEM-'visible void space to underline that this description refers to samples analysed in laboratory. Fr the case of bedding-parallel fractures, we discuss their origin in section 3.2, second parargraph.

Lines 61-64 – There has been a renewed interest in looking at failure in shales at different orientations see the following papers:
Nejati, A. Aminzadeh, F. Amann, M.O. Saar, T. Driesner (2020) Mode I fracture growth in anisotropic rocks: Theory and experiment. Int J Solids Struct, 195 pp. 74-90, 10.1016/j.ijsolstr.2020.03.004
Gehne S, Forbes Inskip ND, Benson PM, Meredith PG, Koor N (2020) Fluid-driven tensile fracture and fracture toughness in Nash point shale at elevated pressure. J Geophys Res: Solid Earth 125:1–11. https ://doi.org/10.1029/2019J B018971
Forbes Inskip ND, Meredith PG, Chandler MR, Gudmundsson A (2018) Fracture properties of Nash Point shale as a function of orientation to bedding. J Geophys Res: Solid Earth. https ://doi. org/10.1029/2018J B0159 43
Chandler MR, Meredith PG, Brantut N, Crawford BR (2016) Fracture toughness anisotropy in shale. J Geophys Res: Solid Earth 121:1–24. https ://doi.org/10.1002/2015J B0127 56
Although these studies mainly use unconfined tests, they demonstrate how the fabric, and alignment of grains in shales has a significant effect on fracture propagation and mode of failure. Although I do not think an in depth discussion of these papers is required, they should at least be mentioned here.
→ We thank the reviewer for these references. We cited them in the text as they provide high quality research to the deformation (anisotropy) of clay-rich rocks/shales.

Line 86 – What is the range of confining stresses tested? You quote the strain range (>20%) so you should also quote the stress range.
→ We inserted the stress range used in the references cited.

Line 99 – This should be "gouge" not "gauge"
→ Corrected

Line 101 – I don't think "modelled experimentally" is the best phrase here, perhaps "has been analysed experimentally by use of direct shear tests on samples sheared both parallel......"
→ We changed the sentence according to you suggestion.

Line 128 – Why did you choose to core the material in this orientation? Are your tests on samples in this orientation the most representative of what you would expect on structures found at Mont Terri and elsewhere in the OPA? As you mentioned earlier, and in the references I provide above,

loading orientation is known to be a significant contributor to failure (fracture mode, strength etc) in transversely isotropic rocks. It would be good to show the reasoning for your sample orientation choice here.

→ Here we would like to refer to comment #1 (line 18) above. We certainly agree that the bedding orientation towards loading plays a fundamental role, but at the time this manuscript has been prepared, we have not finished our study on different loading configurations - it is currently in progress and submission will be in 2021/2022 to constrain a broader spectrum of loading configurations.

Line 134 – Similar to my comment above, why did you choose a circumferential displacement rate of 0.08 mm/min. Many failure processes are heavily rate dependent, are these rates relevant to the application of your study? Or is this rate defined in an ISRM suggested method?

→ The test aimed for undrained conditions. i.e. the test needed to be conducted with a deformation rate appropriate for the sample size to fulfill undrained conditions. As suggested in a draft for IRSM (Fairhurst and Hudson, 1999) Amann et al. 2012 adapted the strain rate slightly from 1E-6 1/s as suggested for radial controlled strain deformation to 4,8E-6 1/s, i.e. 0.08 mm/min. We inserted the reference to the text.

Line 139 – This section relates only to sample preparation for the image analysis, and so should be titled as such. Something along the lines of "Sample preparation for image analysis". Furthermore, I would also change section 2.1 to something like "Material description and core sample preparation" or "Material description and sample preparation for mechanical testing" if you want to keep the two types of sample preparation separate. Either that or you could remove lines 126 – 130 from section 2.1 and incorporate it with section 2.3 for a more general sample preparation section, but this should then come before the section on Triaxial testing.

→ We agree and changed the section title according to your seggestions:
2.1 Material description and core sample preparation
2.2 Sample preparation for image analysis and BIB-SEM

Line 153 – Do you really need to use the acronym ROI for regions of interest? I do not think it saves much in the way of words, and in my opinion the use of another acronym is confusing for the reader. You also only use it once in the whole MS.

→ We agree and deleted the acronym.

Line 163 – I would use either sub-horizontal or parallel to bedding, and not both. My preference would be to always use an orientation related to the bedding, so parallel in this case. The reason being that the bedding relates to the fabric of the rock and will remain (relatively) constant, whereas horizontal or vertical can be different depending on how you orientate your sample. Also be consistent then with how you define both fracture sets, i.e.oblique to bedding (fracture set 1) and parallel to bedding (fracture set 2).

→ We see a point in relating the orientation towards the bedding direction and would usually agree. But as the bedding, in this case, is oriented approximately 9° from the horizontal (not an 'ideal' S-sample) it would make things more complicated. Therefore, we measure all angles from horizontal (for clarification we inserted a sketch in Fig.3). We changed all text parts, where possinle, to make this consistent throughout the manuscript.

Line 205 – This should be "saddle reef pores" rather than "raddle reef pores" I think. I have to admit, this is not a term I am familiar with. Do you need to define this, or at least annotate examples in Figure 7?

→ Typo corrected. This term stems from (micro)structural geology and has been used elsewhere, we inserted the citation and inserted an annotation to Fig.6

Lines 211 – 214 – It is not clear to me where these porosities have been measured. Could the sub-areas used in this analysis be marked on Figure 7?

→ We inserted the locations for the porosity measurements in Fig.7 (blue boxes, dashed lines). We also inserted a reference in the text. There is one regions missing on the figure, which was used to calculate the porosity of the deformation band; we decided not to show this image as which would have caused information overload of the figure, it is however located in the deformation band 7(d)

Lines 219 and 224 – I think you have mixed up fracture sets 1 and 2 here. In section 3.1 you define fracture set 1 as the set oblique to bedding while fracture set 2 is parallel to bedding. Please amend to be consistent.

→ That's true. We corrected the false numbering.

Line 226 – Here you go back to defining features from the horizontal rather than bedding. Switching between the two is confusing, so please stick to one or the other. As mentioned previously I think orientating with regards to the bedding is better.

→ We decided to measure the angle with respect to horizontal (see comment above). We are therefore consistent throughout the manuscript.

Line 230 – Further to my previous comment: You now use the bedding direction rather than horizontal. You use both within the same paragraph, please be consistent.

→ We inserted a comment, which states that these fractures are oriented with ca. 9° towards the horizontal, which is in our case the bedding orientation. We feel that expressing the orientation in relation with bedding is needed, as desiccation cracks are prone to develop along the bedding.

Lines 233 – 234 – "In general, the deformation of both the clay-rich matrix and larger quartz, calcite and mica grains is brittle, ductile or a combination", brittle, ductile or a combination covers everything and so this sentence in its current form is rather redundant in my opinion. I understand what you mean, in that you do not just have one type of deformation and that both occur, but maybe then consider re-writing the sentence to something like "We observe both brittle and ductile deformation in the clay-rich matrix and larger quartz, calcite and mica grains, and so deformation is not solely brittle or ductile."

→ As also criticised by the other reviewers, we changed the sentence to express the information is a better sense.
*"is governed by brittle or ductile processes."*
You also list clay-rich matrix, quartz, calcite and mica here, what else is there of significant quantities? Do you not see deformation in the Iron rich minerals and/or feldspars? If not is this because they account for <10% of the rock (from section 2.1.), and therefore could this simply be a sampling bias? I just wonder if you need to specifically list clay-rich matrix, quartz, calcite and mica, or if you could just say that you observe both brittle and ductile deformation. Finally, I personally do not think you need to say a combination of both, as I believe it is covered when saying that you observe brittle and ductile deformation. However, if you feel strongly about keeping that in that is fine.

→ We agree with your point and added pyrite aggregates to the sentence since these present also deformed patterns. Feldspars are rarely observed in our samples; this could be certainly be due to sampling bias and the fact that feldspar is only a minor component. Nevertheless, we would like to keep this sentence as is, and list the mineral component to stress that deformation has been taken place in all mineralogical phases regardless the rheology contrast.

Lines 265 – 268 – Similar to my comments above, the way that the first sentence here is written is a bit redundant as elastic and inelastic deformation covers everything. Again, I understand what you mean in that both occur, but then consider re-writing to something like "We observed that both elastic and inelastic deformation occurred during testing, and that both occurred simultaneously". You mention in line 268 that both occur simultaneously but have you got evidence for this? Also

could pore compression (line 266) not be inelastic rather than just elastic? Or do you consider inelastic pore compression as pore collapse?

→ This sentence, also according to the other reviewers comments, has been changed based on your suggestion. We changed the following sentence as well to make the argumentation more consistent. We can see that plastic and elastic deformation occurs simultaneously as the deviation of the volumetric stress-strain curve starts early (at low axial strain) and is therefore inferred to occur simultaneously. We also state that bulk compression (i.e. clay matrix compaction and pore collapse) is a part of plastic deformation (= inelastic).

*"In our triaxial experiment, we observe that elastic and irreversible inelastic deformation occurred, and that both occurred simultaneously. Elastic deformation took place by deformation of the solid component and of pores – this was inferred to be reversed by unloading the sample. From our SEM investigations, plastic deformation was governed by damage accumulation in deformation bands, but also by homogeneous matrix bulk compaction (cf. Allirot et al., 1977). Bulk compression in clay-rich rocks is associated with clay matrix compaction and pore collapse (Schuck et al., 2020). […]"*

Line 274 – Here you now use angle with respect to the maximum principle (presumably – but need to state this, as in engineering σ1 is commonly the maximum principle tensile stress) compressive stress σ1 in addition to angle from bedding and horizontal as before. I can understand why you may want to use σ1/loading direction here as well, but then you should also indicate which fracture set these structures relate to (Fracture set 1?).

→ We inserted the information to what fracture set we are referring to and added – to be consistent the orientation towards horizontal.

Line 284 – It is interesting that your microstructural analysis suggests that tensile cracks form as obliquely rather than vertically (parallel to the loading direction) orientated cracks. You do not mention this here but could this not be down to the transversely isotropic nature of the material, and that you load the sample oblique to bedding? Literature on the fracture toughness of shales at different angles to bedding suggest that during crack growth there is a competition for a tensile fracture to form parallel to load and parallel to the plane of weakness i.e. the short-transverse orientation. As a result, the actual fracture orientation lies somewhere between the two. Again, see the following:

- Nejati, A. Aminzadeh, F. Amann, M.O. Saar, T. Driesner (2020) Mode I fracture growth in anisotropic rocks: Theory and experiment. Int J Solids Struct, 195 pp. 74-90, 10.1016/j.ijsolstr.2020.03.004
- Forbes Inskip ND, Meredith PG, Chandler MR, Gudmundsson A (2018) Fracture properties of Nash Point shale as a function of orientation to bedding. J Geophys Res: Solid Earth. https ://doi. org/10.1029/2018J B0159 43
- Chandler MR, Meredith PG, Brantut N, Crawford BR (2016) Fracture toughness anisotropy in shale. J Geophys Res: Solid Earth 121:1–24. https ://doi.org/10.1002/2015J B0127 56
  Again, these studies are unconfined but I think that the formation of (apparent) tensile fractures in these studies may explain some of the observations that you have here with regards to the orientation of tensile cracks.

→ We agree and comment on this in the discussion 4.1. See also your main comment #1.

Line 292 – Should be "changed" not "changeed"
→ Typo corrected.

Line 320 – You use the British spelling of characterised here (with an s rather than a z) but use American spelling elsewhere (line 14, 84). Generally you use American spelling throughout, but be consistent.
→ We made a spelling/grammar check and converted all inconsistencies to BE.

347 – You use the term obliquely orientated here, but obliquely orientated to what, bedding, horizontal or σ1? You use all three in the MS, and it may be that the cracks are oblique to all three, but you should state what the cracks are oblique to here.

→ We inserted *"to horizontal"* to be consistent within out MS.

Lines 351 – 352 – Here you state that there are few similarities with naturally deformed OPA. The manuscript and study are really interesting, however as you point out in your abstract important questions remain, particularly in relating data gathered in the lab to that observed in the subsurface. Your statement here suggests that you are not able to apply the results of your work to subsurface processes on a larger scale, which in my opinion undermines the great work that you have done. Although it may be true that your results do not corroborate well with naturally deformed OPA, you should expand on why you think this is, and how then your results are relevant. In the final lines of the MS you briefly suggest what could be done next, but you could expand on this to suggest how this study can better inform what types of experiments should be carried out next to tackle these open questions.

→ We completely agree with this comment and as suggested also by the other reviewer, we changed the text and point out (besides the differences) what similarities we found. Furthermore, we included a paragraph on in-situ excavation induced fractures which show also similarities and the significance to the results found in the study. We additionally comment on the experimental BC and relate them to the in-situ BC and conclude therefore structural difference to other studies. Finally, we shortly summaries what experiments should be carried out and – beyond those BC – further processes may take place in the deformation process, which have not been considered in the analysis of this work. A summary of these points is presented in the conclusions as well, see also your main comment #4.

[revised manuscript text omitted]

Line 521 - 522 – change "was conducted circumferential-displacement-controlled..." to "was conducted using a circumferential-displacement-control rate of 0.08 mm/min...".
→ We changed this sentence in incorporation your and the other reviewers suggestion.
*"The strain-controlled test (circumferential-displacement-control rate of 0.08 mm/min) was performed at 4 MPa […]"*
Figure 3 right – The axis is labelled "angle" but what angle is this referring to? It is not made clear in the figure caption and given that the horizontal and the bedding direction are within 10° of each other it isn't immediately obvious which angle you are referring to. Please re label the axis "fracture angle to bedding/horizontal (as appropriate)"
→ We inserted a sketch in Fig.3 illustrating the definition of angles used in our MS. We refer in the text – if not explicitly mentioned – always to the definition as shown here.

Lines 536 – 537 – Here you just say "the oblique orientated fracture set" If you compare both fracture sets to the horizontal (which you do in line 537), they are both oblique i.e. not parallel or perpendicular to the horizontal. This also goes back to my previous comments of use one frame of reference only, bedding or horizontal, throughout. Bedding would be my preference.
→ We agree that both fracture sets are oblique and we deleted "oblique" in line 536-537 and added the missing information of what fracture set we are referring to. However, we would like to stick to our definition for angles as explained above.

Figure 4 – Is Figure 4 (b) an inset from Figure 4 (a)? If so indicate where it is from. Also put a scale in Figures (c) and (d)
→ No 4(b) is not an inset from Fig.4(a). Scales for (c) and (d) were added.

Figure 6 – Should the text in Figure 6 (a) be "Bent mica" rather than "Bend mica", this is also true for the figure caption and line 195 in the main text.
→ All mentioned text passages were changed.

Line 565 – Here you reference (a, a', a") but only a and a' are shown in the figure. Also this may seem pedantic, but in the figure you use the prime symbol whereas you use the apostrophe in the figure caption, this should be consistent.
→ a" was deleted (remnant from a former version of the manuscript).  As the font used in the figures is Arial (as suggested by EGU Solid Earth guideline), the two symbols appear the same unfortunately (as opposed to the text where Time New Roman is used as suggested by SE).

Figure 9 – I really like this figure, it explains nicely the features that you observe in your samples and describes the processes that have taken place. You also show an example of a saddle reef pore, but is there a 'real life' example that you could show in figure 7?

→ We inserted a label in Fig.6 to show a 'real life' example of a saddle-reef-pore.